# Seeing Without Understanding: Disentangling Perception, Reasoning, and Simulation in VLM Gameplay

**Dingyang Jin** [1]   **Jiawei He** [1]   **Calvin Lo** [1]   **Steven Hu** [1]   **Ryan Rad** [1]

## Abstract

While Vision-Language Models (VLMs) excel on static visual benchmarks, they consistently underperform in game-based reasoning, yet existing evaluations conflate failures in perception, rule comprehension, and reasoning. We propose a two-stage diagnostic framework that decomposes VLM performance into testable components: controlled perception tests isolating visual encoding, and a diagnostic matrix with a six-level rule complexity ladder evaluated in both explicit verification and predictive simulation modes. Experimenting with six state-of-the-art VLMs reveals three failure patterns: (1) coordinated spatial drift, where off-by-one localization errors among adjacent pieces share the same shift direction at $1.5$–$1.9\times$ the rate expected under spatial independence; (2) perception-reasoning dissociation, where models correctly verify board states but fail to apply rules—at complex constraint levels, perception remains relatively stable while reasoning accuracy plummets, with even the best-performing model capped at $75\%$; and (3) a simulation gap, with performance dropping by up to 27 points when predicting future states versus verifying observed outcomes. These limitations persist across model scales and are not resolved by scaling, text-only input, or structured prompting. Code and data are available at https://github.com/chillibeaver/PRS-Diag.

## 1. Introduction

Vision-Language Models (VLMs) have achieved remarkable performance on static visual benchmarks, from image cap-

tioning to visual question answering (Radford et al., 2021; Liu et al., 2024). However, recent research highlights persistent limitations in interactive contexts, including deficits in spatial reasoning (Chen et al., 2024), difficulties with long-term planning (Kambhampati et al., 2024), and challenges in maintaining temporal context (Wang et al., 2025b). Games provide a natural testbed for evaluating these capabilities. Importantly, we do not view games as an end application, but as environments with fully specified rules, unambiguous state transitions, and precise ground truth (Cobbe et al., 2021; Hafner et al., 2024). Insights obtained in this setting transfer to structured visual domains such as document understanding, robotic manipulation, user-interface automation, and visual planning, where small perceptual errors can similarly propagate into systematic decision failures.

Recent game-based benchmarks have revealed that VLMs consistently underperform in such environments, often failing even on simple games or game states (Wang et al., 2025a; Ren et al., 2025; Zheng et al., 2025). However, existing evaluations predominantly focus on aggregate performance metrics and rarely provide fine-grained diagnostics that isolate distinct failure modes. A critical gap is error attribution: when VLMs underperform, existing benchmarks do not distinguish whether failures originate from perceptual deficits, rule misunderstanding, or reasoning errors. This work addresses this gap by decomposing VLM failures into independently testable components through controlled experiments.

This work makes three contributions. We design a controlled diagnostic framework that disentangles perception, rule application, and simulation in VLM gameplay, enabling precise attribution of multimodal failures. We also quantify spatially coordinated localization drift in VLM perception, showing that visual errors propagate as structured regional displacements rather than independent noise. Finally, a matched-sample comparison of verification versus prediction reveals a persistent simulation gap that neither scaling, text-only input, nor structured prompting resolves.

---

[1]Northeastern University, Vancouver, BC, Canada. Correspondence to: Dingyang Jin <jin.dingy@northeastern.edu>, Ryan Rad <r.rad@northeastern.edu>.

*Proceedings of the $43^{rd}$ International Conference on Machine Learning*, Seoul, South Korea. PMLR 306, 2026. Copyright 2026 by the author(s).

## 2. Related Work

### 2.1. Game-based VLM Evaluation and Observed Limitations

Game-based benchmarks have emerged as a key methodology for assessing VLM capabilities, spanning turn-based games (Wang et al., 2025a), grid-based puzzles (Ren et al., 2025), and continuous environments (Zheng et al., 2025). These benchmarks evaluate perception, spatial reasoning, and decision-making through metrics such as game completion rates, cell-level accuracy, and ELO ratings.

Empirical results consistently reveal severe limitations. VLMs achieve less than 80% completion even on easy puzzles (Ren et al., 2025), perform at or below random baselines on rule-comprehension tasks (Wang et al., 2025a), and show near-complete failure at object tracking in dynamic games (Zheng et al., 2025). When failure modes have been investigated, visual perception errors emerge as the dominant bottleneck—accounting for over 50% of failures in multiple studies (Zheng et al., 2025; Zhang et al., 2024)—with localization errors being particularly prevalent while element identification remains relatively intact (Ren et al., 2025; Zhang et al., 2024). These findings align with broader evidence that VLMs struggle with basic spatial tasks even outside game settings (Rahmanzadehgervi et al., 2024), and that fine-grained perception remains a primary bottleneck in abstract visual reasoning (Yan et al., 2025). Beyond perception, models exhibit anchoring bias (Zheng et al., 2025), and a tendency to mimic strategic language without genuine reasoning (Wang et al., 2025a).

### 2.2. Diagnostic Approaches

Recent work has introduced methods for decomposing VLM failures. Unit test frameworks isolate assessment of subtasks such as positioning and tracking under simplified conditions (Zheng et al., 2025). Text-only versus multimodal comparisons reveal that VLMs generally perform better without visual input (Zheng et al., 2025; Ren et al., 2025; Chen et al., 2025), highlighting weaknesses in multimodal integration. Visual Q&A at multiple scales enables evaluation of perceptual understanding at both step and cell levels (Ren et al., 2025). Just et al. (Just et al., 2025) propose decoupled training to independently improve visual extraction and logical consistency, complementing the diagnostic perspective of this work.

However, these diagnostic techniques remain limited in scope and application. Failure analysis often emerges as a secondary byproduct of performance evaluation, conducted through post-hoc review of game logs rather than targeted experimentation. Existing approaches do not gate reasoning on verified perception, so a rule-following error may stem from correct reasoning applied to misperceived game states,

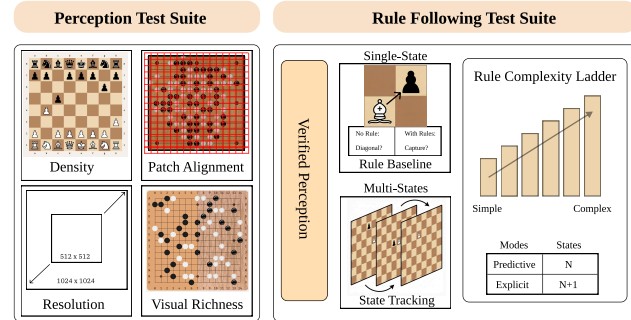

*Figure 1.* **Two-Stage Diagnostic Framework.** Stage 1 isolates VLM performance on perception accuracy. Stage 2 evaluates rule following via a diagnostic matrix, and a six-level complexity ladder with Explicit and Predictive testing modes for fine-grained capability analysis.

or from faulty reasoning despite accurate perception—two fundamentally different failures that remain conflated. PuzzleVQA (Chia et al., 2024) is an exception in that it separates perception from reasoning, but relies on oracle hints that bypass the model's own visual pipeline. Consequently, existing studies offer limited insight into the underlying mechanisms of failure.

In contrast, we treat failure attribution as the primary design objective rather than a secondary byproduct, explicitly isolating perception from reasoning to enable targeted diagnosis.

## 3. Methodology

### 3.1. Overview

Our diagnostic framework decomposes VLM performance into two independently testable stages: perception and rule-following. By evaluating each stage in isolation, we can determine whether a failure originates from visual perception or from reasoning over correctly perceived states. We hypothesize that errors cascade through the processing pipeline: *perception errors propagate to language processing, constraining spatial reasoning and rule application.* Prior work shows VLMs parse chess boards with only ∼55% accuracy (Wang et al., 2025a), imposing a perceptual ceiling on downstream reasoning.

This motivates two design principles: (1) accurate perception is a prerequisite for visual game understanding, making perception diagnosis foundational; (2) reasoning must be assessed only on verified perceptions to correctly attribute failures. Figure 1 illustrates our test suite.

### 3.2. Experimental Setup

**Games.** We evaluate perception on chess ($8 \times 8$) and Gomoku ($15 \times 15$), and rule-following on chess and Xiangqi. Chess provides diverse piece types with complex

movement rules. Gomoku's larger grid and binary piece encoding stress spatial localization while isolating localization errors from piece-type confusion. For rule-following, we substitute Xiangqi for Gomoku—Gomoku's rules are too simple for multi-level evaluation, whereas Xiangqi's distinct rule set tests cross-domain generalization.

**Models.** We evaluate six VLMs: Qwen3-VL (8B, 30B, 235B), GPT-5.2, GLM-4.1V 9B, and Gemma-3 27B. All models except Gemma-3 use native thinking mode. See Appendix A.1 for detailed configurations.

**Evaluation Protocol.** For perception tests, models output board states as numeric matrices compared against ground truth. We report *piece accuracy*—the proportion of occupied cells correctly identified—rather than overall accuracy, which is inflated by trivially correct empty cell predictions. For rule-following tests, we employ two-phase evaluation: (1) verification confirms correct board state perception; (2) reasoning evaluates rule judgment. Only verified cases contribute to reasoning accuracy, isolating visual failures from reasoning failures.

**Sample Size.** Perception tests include 100 samples per subcategory within each test. Rule-following tests include 100 samples per diagnostic matrix quadrant (400 per game) and 50 samples per complexity level per mode (600 per game).

### 3.3. Stage 1: Perception

Perception tests assess VLM ability to accurately perceive game board states under controlled variations in density, patch alignment, image resolution, and visual complexity.

#### 3.3.1. DENSITY TEST

We investigate how piece density affects perception accuracy by generating board states across three density levels (Low, Medium, High) for both chess and Gomoku. Chess positions are generated through simulated gameplay to ensure legal configurations, while Gomoku boards use random placement with balanced color distribution. Detailed density ranges and rendering parameters are provided in Appendix B.

#### 3.3.2. PATCH ALIGNMENT TEST

Modern VLMs divide images into fixed-size patches before encoding. We investigate whether perception accuracy varies when game elements align with versus straddle patch boundaries. For each model, we configure model-specific patch sizes based on officially disclosed specifications and generate images under four alignment conditions by adjusting board position: *Boundary* (offset $= 0$, element centers

at patch boundaries), *Quarter* (offset $= P/4$, centers at $1/4$ into patches), *Center* (offset $= P/2$, centers at patch centers), and *Three-quarter* (offset $= 3P/4$, centers at $3/4$ into patches), where $P$ denotes patch size. We align piece centers for chess and intersection points for Gomoku, as these represent the perceptually relevant elements. Model-specific patch sizes and image resolutions are detailed in Appendix B.3.

#### 3.3.3. RESOLUTION DIVISIBILITY TEST

When image dimensions are not exact multiples of the patch size, preprocessing steps such as resizing, padding, or cropping are required (Varma et al., 2024). We investigate whether non-divisible resolutions degrade perception accuracy compared to cleanly divisible ones. Using the patch sizes established in Section 3.3.2, we test six resolutions per model, split evenly between divisible and non-divisible categories. Non-divisible resolutions are chosen to be close to their divisible counterparts to isolate preprocessing effects from resolution magnitude. Detailed configurations are provided in Appendix B.4.

#### 3.3.4. VISUAL RICHNESS TEST

We compare two rendering styles—*2D* (minimalist geometric shapes with solid colors) and *3D* (realistic textures, lighting, and shadows)—to investigate whether visual complexity aids or hinders perception. Both styles use identical layouts, medium density, and fixed resolution to isolate visual style as the sole variable (see Appendix B.5).

#### 3.3.5. QUANTIFICATION OF STRUCTURAL SHIFT

Beyond aggregate piece accuracy, we investigate the spatial structure of localization errors. Preliminary inspection revealed that adjacent pieces often shift together (Figure 2), suggesting spatially coordinated rather than independent errors. To quantify this, we aggregate data from all four perception tests (density, patch alignment, resolution, and visual richness). To ensure balanced representation across tests with varying sample sizes, we employ two-level weighted averaging: first computing metrics within each test category, then averaging across categories with equal weights.

Specifically, we first pair predictions with ground truth using Hungarian matching (Kuhn, 1955) with Chebyshev distance $\leq 1$, recording the shift direction $d(p) \in D$ for each off-by-one match, where $D$ denotes the set of eight possible directions. Let $S$ denote all shifted pieces. We compute two metrics.

The *dominance ratio* measures directional bias:

$$R_{\mathrm{dom}} = \frac{\max_{d \in D} |\{p \in S : d(p) = d\}|}{|S|} \tag{1}$$

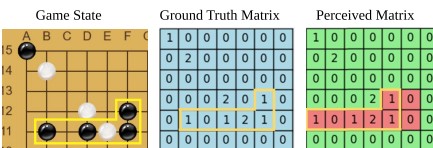

*Figure 2.* **Structural shift example.** Adjacent pieces (yellow box) are displaced one cell together.

Under uniformly random errors, the expected value is $1/8 = 12.5\%$.

However, directional bias alone could arise from independent per-piece tendencies rather than spatially coordinated errors. To test whether adjacent pieces shift together, we compute the *adjacent same-direction ratio*:

$$R_{\text{adj}} = \frac{|\{(p_i, p_j) \in \mathcal{A} : d(p_i) = d(p_j)\}|}{|\mathcal{A}|} \quad (2)$$

where $\mathcal{A}$ denotes pairs of shifted pieces that were adjacent in ground-truth positions (8-connectivity). Since direction distributions may be non-uniform, even independent shifts could yield elevated same-direction rates. To control for this, we construct a Monte Carlo baseline (200 simulations) by randomly permuting the shift directions among piece positions while preserving the total count of each direction—representing the expected ratio under spatial independence. Ratios substantially exceeding this baseline indicate that spatial adjacency increases the probability of same-direction shifts, providing evidence of structural errors where localized board regions are perceptually displaced as a unit.

### 3.4. Stage 2: Rule Following

Rule-following tests assess VLM ability to apply game rules correctly to perceived board states, from basic spatial judgments to complex multi-condition temporal reasoning.

All tests in this stage follow a two-phase evaluation: (1) *Verification Phase* confirms correct board state recognition; (2) *Reasoning Phase* evaluates rule judgment accuracy. This separation isolates visual recognition failures from reasoning failures.

#### 3.4.1. DIAGNOSTIC MATRIX

As shown in Table 1, we organize tests along two orthogonal dimensions: **input modality** (single-state vs. multi-state) and **knowledge requirement** (rule-free vs. rule-based), yielding four diagnostic quadrants.

**Single-State Tests**   Rule-free tests assess spatial reasoning capabilities (e.g., collinearity, relative positions, path clarity), while rule-based tests require movement pattern validation and path blocking detection for specific piece types.

**Multi-State Tests**   Rule-free tests assess state tracking capabilities (e.g., movement detection, sequence ordering), while rule-based tests require history-dependent judgments such as en passant and castling legality. Test cases are designed to enforce multi-state integration—sequences with identical final positions but different histories yield opposite legality judgments (see Figure 16 in Appendix C.1.2).

#### 3.4.2. RULE COMPLEXITY LADDER

While the diagnostic matrix provides a coarse categorization of task types, finer-grained analysis of rule application requires systematic variation along a difficulty gradient. We therefore implemented a six-level complexity ladder that spans a range of rule-based tasks, from basic pattern recognition to advanced temporal reasoning, as shown in Table 2.

Levels progress from single-condition pattern recognition (L1) through multi-condition spatial and temporal verification (L2–L4) to full constraint integration (L5–L6). This gradient provides a controlled framework for examining how perception and reasoning capabilities diverge as task demands increase.

**Two Testing Modes**   Each level employs two complementary modes:

- *Explicit Mode*: Shows N+1 states including the operation result; asks "Is this completed operation legal?" Tests retrospective verification.

- *Predictive Mode*: Shows N historical states; asks "Can this move be made?" Tests forward reasoning.

For example, in a capture scenario, Explicit Mode presents the complete sequence from the initial position through to the captured state, then asks whether the completed capture was legal. Predictive Mode presents only the pre-capture states and asks whether the capturing piece can execute the capture. Both modes test identical rule knowledge but impose different cognitive demands: verification of observed outcomes versus prediction of unobserved outcomes. Comparing performance across modes quantifies this gap and reveals how it evolves with task complexity.

*Table 1.* Diagnostic matrix for rule-following evaluation.

|  | **Single-State** | **Multi-State** |
|---|---|---|
| **Rule-Free** | Spatial Relation | State Tracking |
| **Rule-Based** | Static Constraints | Temporal Constraints |

*Table 2.* Rule Complexity Ladder

| Level | Cognitive Ability Tested | Conditions |
|-------|--------------------------|------------|
| L1 | Movement Pattern Recognition | 1 |
| L2 | Conditional Movement Rules | 2–3 |
| L3 | Path & Obstruction Detection | 2–3 |
| L4 | Global Legality Constraints | 3–4 |
| L5 | Multi-Condition Integration | 4–5 |
| L6 | Advanced Temporal Reasoning | 5–6 |

*Table 3.* Density test: Piece perception accuracy (%) across conditions.

| Model | Chess | | | Gomoku | | |
|-------|-------|--------|------|--------|--------|------|
| | Low | Medium | High | Low | Medium | High |
| Gemma-3 27B | 19.11 | 28.90 | 61.67 | 4.55 | 7.63 | 10.29 |
| GLM-4.1V 9B | 65.48 | 75.05 | 75.37 | 21.59 | 33.24 | 47.63 |
| GPT-5.2 | **97.14** | **96.51** | 99.66 | 54.83 | 70.26 | 79.75 |
| Qwen3-VL 8B | 79.34 | 79.22 | 96.65 | 46.68 | 62.40 | 69.26 |
| Qwen3-VL 30B | 88.33 | 87.89 | 92.41 | 52.01 | 61.53 | 68.95 |
| Qwen3-VL 235B | 95.09 | 96.24 | **99.84** | **71.86** | **74.69** | **84.20** |

# 4. Experimental Results & Discussion

## 4.1. Perception Results

### 4.1.1. DENSITY TEST

Table 3 summarizes piece perception accuracy across density levels. GPT-5.2 and Qwen3-VL 235B achieve the highest accuracy, while Gemma-3 27B performs poorly on both games. Within the Qwen3-VL family, only the 235B variant consistently outperforms smaller variants; the 8B and 30B models show inconsistent ordering, suggesting that intermediate scaling does not reliably improve perception.

**Density-Accuracy Trend** Figure 3 visualizes per-sample accuracy against piece count. In chess, accuracy generally increases with density. However, this trend is partly confounded by positional canonicity: chess boards are generated through simulated gameplay , so high-density boards disproportionately represent early-game positions familiar from training data. A controlled experiment using randomized placements (Appendix D.1.1) shows that the strong monotonic trend weakens, but high density does not degrade accuracy for any model and continues to benefit some (e.g., Gemma-3 27B, Qwen3-VL 30B). Gomoku boards use random placement by design, providing a cleaner test free from this confound. Here, accuracy increases consistently with density across most models. The original chess trend partly reflects training data familiarity, but higher feature density may still aid spatial localization.

**Prediction Bias Analysis** Beyond aggregate accuracy, we examine the structure of model errors. Treating each cell's occupancy as binary classification, we ask: when models make errors, do they tend toward false negatives (predicting empty when a piece exists) or false positives (predicting

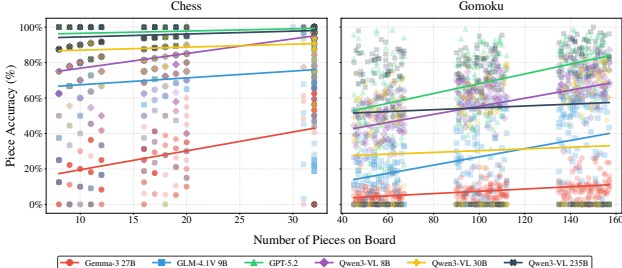

*Figure 3.* **Piece perception accuracy vs. board density.** Each point represents one test sample; lines show linear trends. Left: Chess. Right: Gomoku. The chess trend is partly confounded by positional canonicity.

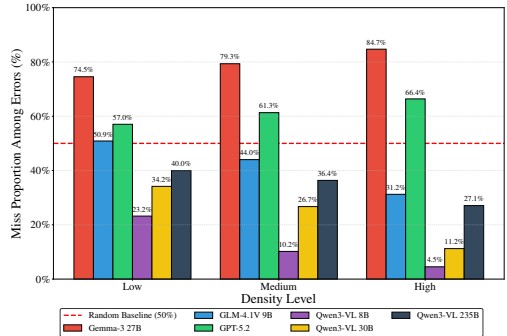

*Figure 4.* **Model bias in piece existence prediction.** Miss proportion = FN/(FN + FP) measures the fraction of errors that are misses vs. false alarms. Dashed line: 50% random baseline. Above 50%: conservative bias (prefers predicting empty); below 50%: aggressive bias (prefers predicting piece).

a piece where none exists)? We define *miss proportion* as FN/(FN + FP). A random predictor yields exactly 50% regardless of board density, serving as our baseline.

Figure 4 shows miss proportion across density levels. Two behavioral patterns emerge. GPT-5.2 and Gemma-3 27B exhibit *conservative bias*, consistently exceeding the 50% baseline—when uncertain, these models tend to predict "empty." Gemma-3 shows extreme conservatism (75–85%), intensifying at higher densities. In contrast, the Qwen3-VL family and GLM-4.1V display *aggressive bias*, with miss proportions well below 50% that decrease further at high density—these models prefer predicting pieces over empty cells, even at the cost of more false alarms.

### 4.1.2. PATCH ALIGNMENT TEST

Across all models and both games, accuracy variations remain small: most models fluctuate within 3–5 percentage points across conditions, with no consistent pattern favoring any particular alignment. These results suggest that the evaluated models are robust to whether visual elements fall at patch boundaries, centers, or intermediate positions. See Appendix D.1.2 for detailed results.

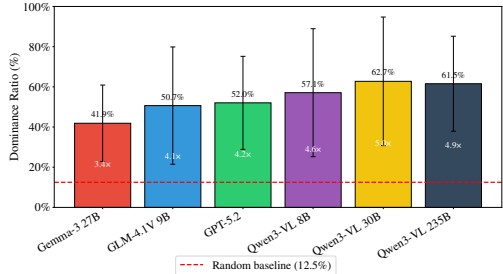

*Figure 5.* **Shift direction dominance ratio.** Dashed line: 12.5% random baseline. All models show strong directional bias (3–5× baseline).

### 4.1.3. RESOLUTION DIVISIBILITY TEST

Resolution divisibility has minimal impact on perception accuracy: divisible and non-divisible resolutions yield differences typically within 1–2 percentage points. However, resolution magnitude matters substantially—on Gomoku, accuracy drops sharply from high to low resolution (e.g., GPT-5.2: 75% → 32%). See Appendix D.1.3 for detailed results.

### 4.1.4. VISUAL RICHNESS TEST

Despite substantial differences in visual complexity between 2D and 3D styles, accuracy differences are modest (typically 1–5 percentage points) and inconsistent in direction—no model exhibits consistent style preference across both games. Visual richness, then, neither systematically aids nor hinders perception. See Appendix D.1.4 for detailed results.

### 4.1.5. STRUCTURAL SHIFT ANALYSIS

**Directional Bias** Figure 5 shows the dominance ratio for each model. All models substantially exceed the 12.5% random baseline, with Gemma-3 27B at the low end averaging 41.9 (3.4× baseline) and Qwen3-VL 30B at the high end reaching 62.7% (5.0× baseline). This confirms that off-by-one errors exhibit strong directional bias across all evaluated models.

**Structural Coordination** Figure 6 compares the adjacent same-direction ratio against Monte Carlo baselines. All models exhibit actual ratios 1.5–1.9× higher than baseline means. GPT-5.2 shows the strongest structural coordination: 54.2% actual vs. 28.9% baseline (1.9×), with the actual ratio exceeding the Monte Carlo 95th percentile (46.7%). Qwen3-VL 235B demonstrates similarly strong effects at 50.9% vs. 33.8% baseline (1.5×), approaching the 95th percentile threshold. The Qwen3-VL 8B and 30B variants show consistent ratios of 37.4% and 38.1% respectively (both 1.5–1.6× baseline). Even GLM-4.1V 9B and Gemma-3 27B, despite their lower overall perception accuracy, exhibit ratios of 27.1% and 28.5%—still 1.5× and 1.7× their respective

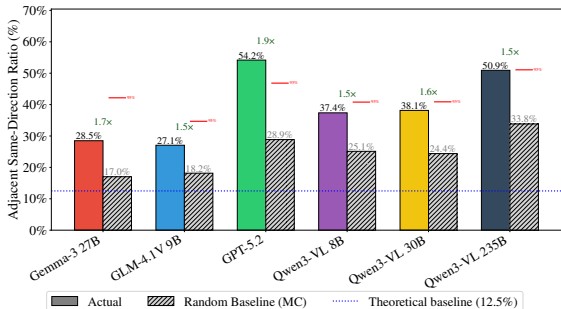

*Figure 6.* **Adjacent same-direction ratio vs. Monte Carlo baseline.** Ratios exceeding baseline indicate spatially coordinated shifts.

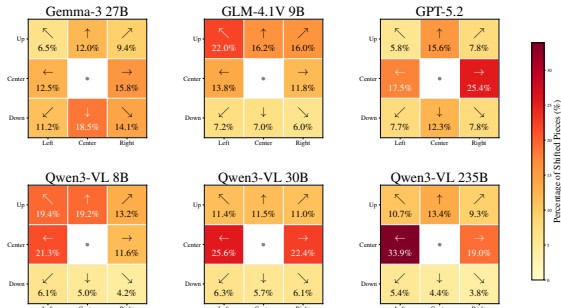

*Figure 7.* **Shift direction preference by model.** Random expectation: 12.5% per direction.

baselines. The consistency of this effect across architectures and scales confirms that spatial adjacency increases same-direction probability beyond what direction bias alone predicts—evidence that localized regions shift as a unit.

**Direction Preferences** Figure 7 reveals model-specific directional biases. Under uniform random shifts, each direction would occur 12.5% of the time. The Qwen3-VL family shows leftward bias (1.7–2.7× baseline), while GPT-5.2 favors rightward shifts (2.0× baseline). These consistent preferences suggest systematic biases in spatial encoding rather than random errors.

The spatially coordinated nature of these errors, combined with the robustness to tokenization boundaries observed in Section 3.3.2, rules out the tokenization process as the source of coordinated drift.

### 4.2. Rule Following Results

Table 4 reports two metrics: $\text{Acc}_v$ (verified accuracy), the reasoning accuracy computed exclusively over cases where perception was verified correct, and Ver (verification rate), the proportion of test cases where the model perfectly perceived the board state. All models demonstrated sufficient board recognition for Diagnostic Matrix tests. However, for Rule Complexity Ladder tests, only the Qwen3-VL family and GPT-5.2 achieved adequate perception rates. The remaining models, Gemma-3 27B and GLM-4.1V 9B, failed

*Table 4.* Diagnostic Matrix results. Each cell: $\text{Acc}_v$ (verified accuracy) / Ver (verification rate) (%). Averaged over Chess and Xiangqi.

| Model | Single-State | | Multi-State | |
|---|---|---|---|---|
| | w/o Rule | w/ Rule | w/o Rule | w/ Rule |
| Gemma-3 27B | 71.7/50.5 | 55.5/57.0 | 93.7/49.5 | 49.9/37.5 |
| GLM-4.1V 9B | 85.0/90.0 | 48.4/87.5 | 90.7/48.0 | 65.5/31.0 |
| GPT-5.2 | 98.5/99.5 | 92.5/97.5 | 100.0/98.0 | 92.5/99.5 |
| Qwen3-VL 235B | 98.5/86.0 | 99.0/94.0 | 100.0/85.0 | 91.7/91.0 |
| Qwen3-VL 30B | 96.0/87.5 | 90.0/78.0 | 100.0/73.5 | 82.2/76.0 |
| Qwen3-VL 8B | 94.0/86.0 | 78.1/83.0 | 92.2/58.0 | 80.9/69.0 |

at the verification stage, exhibiting near-total perception failure at higher complexity levels—<1% for Gemma-3 and <3% for GLM-4.1V at Level 6—precluding meaningful assessment (see Appendix D.2.1 for detailed verification rates).

### 4.2.1. BASELINE CAPABILITY VERIFICATION

All models achieved 90–100% $\text{Acc}_v$ on Multi-State rule-free tasks, with most showing similarly high accuracy on Single-State rule-free tasks (Table 4), establishing sufficient baseline for sequential processing and spatial reasoning. When domain rules were introduced, performance degraded substantially—with Multi-State showing larger drops than Single-State for most models—indicating the difficulty lies in rule integration itself, not basic perceptual or sequential processing.

### 4.2.2. PERCEPTION-REASONING DISSOCIATION

The rule complexity ladder experiments reveal a significant dissociation between visual perception and logical reasoning as shown in Figure 8. Models demonstrate high verification accuracy—correctly identifying piece positions, move sequences, and board states—yet fail to apply this perceived information during rule evaluation.

If perception and reasoning were tightly coupled, accuracy given verified perception should approach 100%—correct perception would guarantee correct reasoning. At Level 1, this expectation holds: Qwen3-VL 235B achieves 99% reasoning accuracy given verified perception. However, as task complexity increases, reasoning accuracy declines substantially even when perception remains accurate. By Level 4, the dissociation becomes severe: Qwen3-VL 235B maintains 90% perception accuracy, yet reasoning accuracy given verified perception drops to only 56%. The divergence is quantifiable: across L1–L5, Qwen3-VL 235B's perception verification rate varies by only 8 percentage points (90–98%), while reasoning accuracy given verified perception drops by 43 points (99% → 56%). GPT-5.2 shows a similar pattern.

A clear example occurs in en passant legality tests. Models correctly perceive that the opponent's pawn advanced

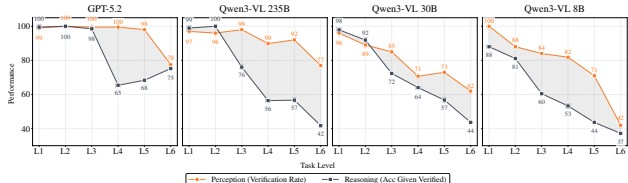

*Figure 8.* Perception-Reasoning Dissociation (Predictive Mode). Perception verification rate (orange) and reasoning accuracy given verified perception (dark blue) across task complexity levels. The shaded region highlights the divergence between stable perception and declining reasoning—demonstrating that reasoning failures occur independently of perceptual errors.

only one square to stand shoulder-to-shoulder with the player's pawn, yet still conclude that en passant capture is legal—despite the rule requiring a two-square advance from the starting position. The model accurately identifies the pawn positions and movement history but fails to apply a trivial check: did the pawn move one square or two? Visual symbols are correctly encoded but fail to transform into logical constraints. See Appendix D.2.1 for additional examples.

This dissociation emerges consistently once task complexity exceeds basic pattern recognition, and persists across model scales. While increasing parameters improves reasoning accuracy at simple levels (L1–L2), returns diminish substantially at higher complexity: at Level 4, even GPT-5.2 achieves only 65% reasoning accuracy given verified perception.

### 4.2.3. SIMULATION GAP: MODELS EXHIBIT WEAK INTERNAL SIMULATION CAPABILITIES

The comparison between Explicit and Predictive modes reveals a persistent limitation in VLM reasoning. While both modes test identical rule knowledge, Predictive mode requires models to internally simulate unobserved outcomes, whereas Explicit mode presents the completed operation for verification. If models possessed strong internal reasoning capabilities, performance should remain consistent across both modes—yet we observe consistent divergence.

A potential confound is that the two modes exhibit different verification rates (Appendix Table 13): meaning their $\text{Acc}_v$ is computed on different sample subsets. To control for this selection bias, we compute accuracy only on cases where both modes passed verification:

$$\mathcal{S}_{\text{valid}} = \{x \in \mathcal{C} \mid V_E(x)\} \cap \{x \in \mathcal{C} \mid V_P(x)\} \quad (3)$$

where $\mathcal{C}$ denotes all test cases and $V_E(x)$, $V_P(x)$ indicate verification success in each mode.

Figure 9 presents the aggregated results on this matched subset. At Levels 1–2, both modes achieve near-identical accuracy (∼95%), indicating that simple movement patterns

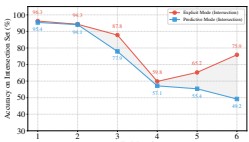

*Figure 9.* Matched-sample analysis: Explicit vs. Predictive mode (aggregated). Accuracy computed on $\mathcal{S}_{\text{valid}}$, controlling for selection bias. The shaded area highlights the widening gap.

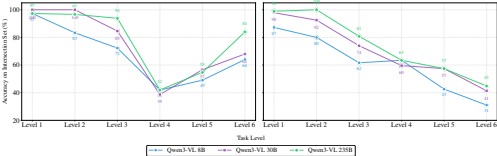

*Figure 10.* Model Scaling Trend Analysis: Explicit vs. Predictive Mode (Given Verified). Left: Explicit mode performance; Right: Predictive mode performance across task levels

can be evaluated without significant simulation overhead. At Level 3, a substantial gap emerges: Explicit mode achieves 87.8% while Predictive mode drops to 77.9%. This gap widens at higher complexity levels, reaching 26.7 percentage points at Level 6 (75.9% vs 49.2%). The persistence of this gap on identical samples confirms that the performance difference is not an artifact of sample selection.

Per-model analysis reveals that the simulation gap emerges consistently across all evaluated models, though with varying severity; see Appendix D.2.1 for details.

### 4.2.4. SCALING EFFECTS: ASYMMETRIC RETURNS ACROSS TASK TYPES

Following a similar approach to Section 4.2.3 to avoid selection bias, we compute accuracy on samples where all three Qwen3-VL variants (8B, 30B, 235B) passed verification within each mode (Figure 10). The results reveal highly inconsistent scaling patterns. In Explicit mode, the 235B model consistently underperforms the 30B variant at Levels 1, 2, and 5. At Level 4, all three models converge to nearly identical accuracy across both modes, with differences limited to $2-3$ percentage points. Rather than monotonic improvement with scale, an $8\times$ increase in parameters from 30B to 235B yields no systematic advantage; the 235B model performs comparably to or strictly worse than the 30B variant across multiple complexity levels. The performance inversions are not statistical artifacts: the primary bottleneck lies not in model capacity, and scaling alone does not address it.

### 4.2.5. ABLATION STUDIES: DISENTANGLING MODALITY AND PROMPTING EFFECTS

The preceding findings (perception-reasoning dissociation, simulation gap, asymmetric scaling) are all observed under image-based, zero-shot evaluation. Two natural questions

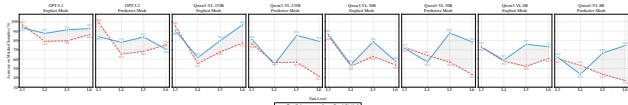

*Figure 11.* Text-only (FEN) vs. image-based (verified) reasoning accuracy on matched samples (L3–L6).

arise: (1) do reasoning failures stem from the multimodal processing pipeline specifically, or from general reasoning limitations shared with the underlying LLM? (2) can structured prompting strategies close the simulation gap? We address these through text-only and Chain-of-Thought (CoT) ablations.

**Text-Only Baseline.** We replace image input with FEN strings for both Chess and Xiangqi, keeping the reasoning question identical. To ensure a fair comparison, accuracy is computed only on the subset of cases where the image-based run passed verification. We focus on L3–L6, where reasoning failures are prevalent; L1–L2 are near ceiling for all models. Figure 11 presents per-model results.

At lower complexity (L3), image-based and text-only reasoning perform comparably, with image-based sometimes ahead (e.g., GPT-5.2 Predictive: image 98.5% vs. text 83.9%). As complexity increases, text-only increasingly outperforms the image pathway, particularly in Predictive mode (e.g., Qwen3-VL 235B Predictive L6: text 79.0% vs. image 41.7%). Per-game breakdowns (Figure 21 in Appendix) confirm that these patterns hold consistently across Chess and Xiangqi.

These results reveal a dual bottleneck. First, text-only reasoning itself degrades with complexity, confirming that general reasoning limitations of the base LLM contribute to the observed failures. Second, the image-based pathway incurs a distinct, quantifiable additional cost in most conditions. This multimodal transformation overhead is especially severe in high-complexity Predictive mode. This overhead is not a product of perception errors, since accuracy is computed exclusively on verified cases; rather, the visual-to-symbolic transformation process itself introduces an independent reasoning cost that compounds with task complexity.

**Chain-of-Thought Ablation.** All models except Gemma-3 already operate with native thinking mode enabled. To test whether additional explicit CoT helps, we augment the original prompts with a structured scratchpad requiring models to sequentially perform coordinate identification, rule retrieval, path enumeration, and constraint checking before reaching a final judgment (full prompt in Appendix D.2.2). The CoT ablation retains both image input and verification gating from the original setup; the only modification is the appended scratchpad. We evaluate all four models on L3–L6

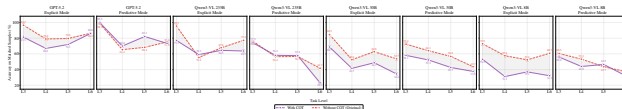

*Figure 12.* CoT ablation vs. original prompting on matched samples (L3–L6).

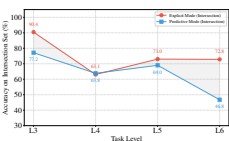

*Figure 13.* Simulation gap under CoT prompting: Explicit vs. Predictive mode on matched samples (L3–L6).

matched samples. Figure 12 presents the results.

CoT does not reliably improve performance and frequently degrades it. In Explicit mode, CoT underperforms the original across nearly all models and levels, with smaller models hit hardest (e.g., Qwen3-VL 8B Explicit L4 drops from 57.6% to 31.1%). Even GPT-5.2 sees Explicit L3 decline from 96.5% to 81.3%. In Predictive mode, CoT yields occasional gains for GPT-5.2 at intermediate levels (e.g., L5: 82.1% vs. 68.4%) but produces sharp decline for Qwen3-VL 235B at L6 (23.1% vs. 41.7%). Per-game breakdowns (Figure 22 in Appendix) confirm that these patterns hold consistently across Chess and Xiangqi.

Importantly, the simulation gap persists under CoT. Figure 13 presents the aggregated matched-sample analysis under CoT prompting. At L6, Explicit mode achieves 72.8% while Predictive mode drops to 46.8%, a gap of 26.0 percentage points, comparable to the 26.7-point gap observed without CoT (Figure 9). These results confirm that the simulation gap is a structural limitation rather than a prompting artifact.

## 5. Conclusion

**Perception Findings**  While our experiments are conducted in game environments, our objective is to use games as controlled probes for understanding multimodal reasoning. The observed failure modes reflect general limitations in how VLMs integrate visual encoding with symbolic and sequential reasoning. Our tests reveal that while VLMs are robust to patch alignment and resolution divisibility, they exhibit a systematic structural shift: adjacent pieces shift together at rates $1.5 - 1.9\times$ higher than expected under spatial independence. This spatially coordinated drift suggests that visual encoding failures are not random but systematic, with localized regions being perceptually displaced as a unit and each model exhibiting distinct directional preferences. Such bias has implications for document understanding and robotic manipulation. High density does not consistently

degrade perception accuracy in either game and in some cases improves it, suggesting that higher feature density may aid rather than hinder spatial localization.

**Rule-Following Findings**  We identify a severe perception-reasoning dissociation: models accurately verify board states yet fail to apply this information during rule evaluation. As task complexity increases, reasoning accuracy drops sharply while perception remains stable. Furthermore, we characterize a simulation gap, where models can perform retrospective verification of observed states but fail at prospective simulation of unobserved outcomes. These limitations persist across model scales. As confirmed by text-only and CoT ablations, the observed failures reflect both general reasoning limitations of the base LLM and a distinct multimodal transformation overhead, neither of which is resolved by scaling or structured prompting.

**Limitations and Future Work**  Our scope is limited to grid-based spatial structures, where perception and reasoning can be meaningfully decomposed. This assumption does not hold for all visual reasoning tasks, such as perspective taking or embodied spatial reasoning, where perception and reasoning are more tightly intertwined. Future work should expand to non-grid tasks and investigate whether coordinated shifts are a byproduct of patch-based ViT architectures. Mechanistic analysis of internal representations is needed to clarify the conditions under which density aids localization and why complex rules trigger reasoning collapse.

## Acknowledgements

We thank Dr. Hazra Imran for her mentorship, and support throughout this project. We also thank the anonymous reviewers and chairs for their constructive feedback.

## Impact Statement

This paper presents work whose goal is to advance the field of Machine Learning, specifically in understanding the systematic limitations of Vision-Language Models (VLMs). By characterizing the fundamental dissociation between perception and reasoning, our work provides a diagnostic framework for identifying failure modes in VLM-driven decision-making. These insights are critical for the responsible development and auditing of multimodal systems, ensuring that visual verification is reliably coupled with logical rule application. While there are potential societal consequences regarding the safety and reliability of AI systems, we believe our findings primarily serve to improve the transparency and interpretability of VLM failure mechanisms.

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

# A. Experimental Setup

## A.1. Model Configurations

Table 5 summarizes the model configurations used in all experiments. All API calls used default parameters without temperature or token limit specifications.

*Table 5.* Model configurations.

| Model | API Identifier | Thinking |
|-------|----------------|----------|
| Qwen3-VL 8B | qwen3-vl-8b-thinking | Yes |
| Qwen3-VL 30B | qwen/qwen3-vl-30b-a3b-thinking | Yes |
| Qwen3-VL 235B | qwen/qwen3-vl-235b-a22b-thinking | Yes |
| GPT-5.2 | gpt-5.2-2025-12-11 | Yes |
| GLM-4.1V 9B | THUDM/GLM-4.1V-9B-Thinking | Yes |
| Gemma-3 27B | google/gemma-3-27b-it | No |

## A.2. Prompting Strategy

All tests use zero-shot prompting. This section consolidates prompts for both perception and rule-following stages.

### A.2.1. PERCEPTION PROMPTS

Models receive a single image and a text prompt requesting board state extraction in matrix format.

**Chess Perception Prompt**

```
Analyze this chessboard image. Output the
board state as an 8x8 JSON matrix.
Encoding:
- Empty: 0
- White: Pawn=1, Knight=2, Bishop=3,
         Rook=4, Queen=5, King=6
- Black: Pawn=-1, Knight=-2, Bishop=-3,
         Rook=-4, Queen=-5, King=-6
Row 0 = top of image (rank 8),
Row 7 = bottom of image (rank 1).
Output ONLY the matrix, no explanation.
```

**Gomoku Perception Prompt**

```
Analyze this Gomoku board image. Output the
board state as a 15x15 JSON matrix.
Encoding: Empty=0, Black=1, White=2
Row 0 = top of image, Row 14 = bottom.
Output ONLY the matrix, no explanation.
```

### A.2.2. RULE-FOLLOWING PROMPTS

Rule-following tests employ a two-phase evaluation within a single prompt: verification confirms correct board state perception before the reasoning question is posed.

```
Look at these board states carefully.
[Image references]
[Context label]

First, a verification question to make
```

sure you see the states correctly:
```
[Verification question]

For verification, use this format:
- List pieces as:
  [Color] [Piece Type] at [square]
- Separate states with semicolons

Now, the main question:
[Complex Rule Clarifications]
[Rule judgment question]

Answer both questions as:
Verification: [your answer]
Main answer: [yes/no/unknown] or [A/B/C/D]
```

**Xiangqi Piece Mapping Instruction** For Xiangqi tests, an additional instruction is appended before the main question to help models recognize Chinese chess piece symbols:

```
Note:  Please use the following English names for
the pieces shown on the board:
- 帥/將= King
- 仕/士= Advisor
- 相/象= Bishop
- 馬= Knight
- 車= Rook
- 炮/砲= Cannon
- 兵/卒= Pawn

Red pieces have red borders, Black pieces have
dark borders.
Please answer using the format:
"[Color] [PieceName] at [position]",
e.g., "Red Rook at e4".
```

### A.2.3. COMPLEX RULE CLARIFICATIONS

Prior to the main experiments, we conducted rule knowledge verification for all evaluated models. With web search disabled, we prompted each model to explain the complete rules of Chess and Xiangqi, including piece movements, special rules (e.g., castling, en passant, flying general), and win conditions. All six models demonstrated comprehensive and accurate rule knowledge, providing detailed and fully correct explanations.

Despite this confirmed rule proficiency, we include explicit rule descriptions in prompts for test cases involving complex multi-condition rules. This design choice serves experimental rigor: by providing rule specifications directly, we eliminate any potential ambiguity and ensure that observed failures are unambiguously attributable to the inability to apply rules during visual reasoning, rather than rule ignorance or misunderstanding.

**En Passant**

```
Note: En passant capture is legal only when:
(1) the opponent's pawn has just advanced
two squares from its starting position, and
(2) your pawn is on an adjacent file on the
```

```
5th rank (for White) or 4th rank (for
Black).
The capture must be made immediately on
the next move, or the right is forfeited.
```

### Castling

```
Note: Castling is legal only when all of
the following conditions are met:
(1) the King has never moved,
(2) the Rook involved has never moved,
(3) there are no pieces between the
King and the Rook,
(4) the King is not currently in check,
(5) the King does not pass through a
square attacked by an enemy piece, and
(6) the King does not end up in check.
```

### Perpetual Check and Chase

```
Note: In Xiangqi, Perpetual Check and
Perpetual Chase are illegal after 3
consecutive occurrences.
```

### Flying General

```
Note: In Xiangqi, the Flying General rule
states that:
(1) the two Kings cannot face each other
directly on the same file (column) with no
pieces between them,
(2) if a piece is the only blocker between
the two Kings, it cannot move to a position
that would expose the Kings to face each
other, and
(3) this rule applies even if the move is
otherwise legal.
```

### A.3. Output Parsing and Verification

#### A.3.1. PERCEPTION OUTPUT PARSING

Model outputs are parsed using a two-stage strategy: (1) regex matching for complete matrix patterns, (2) fallback extraction of numeric values if the first strategy fails. Outputs that cannot be parsed into valid matrices are marked as parse failures.

#### A.3.2. RULE-FOLLOWING VERIFICATION PROTOCOL

Verification responses are validated through two sequential checks:

**Keyword Presence**   For each piece, three keywords are extracted: position (e.g., `e4`), piece type (e.g., `rook`), and color (e.g., `white`). All keywords must appear in the response (case-insensitive, punctuation removed).

**Color-Piece-Position Association**   Beyond keyword presence, we verify correct associations using regex. For each piece with expected color $c$, type $t$, and position $p$:

1. At least one valid pattern must exist (e.g., "$c\,t\,...\,p$")

2. No invalid pattern with opposite color $\bar{c}$ may exist

This prevents cases where models mention correct keywords with wrong associations (e.g., "White Rook at e4" when ground truth is "Black Rook at e4"). Only verified cases contribute to $\text{Acc}_v$.

## B. Perception Test Details

This section provides implementation details not covered in the main text. For test design rationale and configuration specifications, see Section 3.

### B.1. Board State Generation

**Chess**   Board states are generated through simulated gameplay rather than random piece placement, ensuring all positions are legally reachable. Starting from the initial position, the generator plays random legal moves with a bias toward captures when reducing piece count. This approach guarantees valid piece configurations (e.g., pawns never appear on the first or eighth rank, kings are never in illegal proximity).

**Gomoku**   Board states are generated through random placement with balanced color distribution. Positions are sampled uniformly from unoccupied intersections, with pieces alternating between black and white to maintain approximate color balance.

### B.2. Density Test

Table 6 specifies the density levels used for each game.

*Table 6.* Density level configurations.

| Game | Metric | Low | Medium | High |
|------|--------|-----|--------|------|
| Chess | Piece count | 8–12 | 16–20 | 28–32 |
| Gomoku | Occupancy | 20–30% | 40–50% | 60–70% |

All density test images are rendered at $1024 \times 1024$ resolution with coordinate labels. We report *piece perception accuracy*—the proportion of ground-truth pieces correctly identified at their exact positions—rather than overall cell accuracy, as the latter is inflated by the preponderance of empty cells at lower densities.

### B.3. Patch Alignment Test

**Model-Specific Configurations.**   Different VLM architectures employ different patch sizes and image preprocessing strategies. We configure tests based on officially disclosed specifications from each model's technical report: Qwen3-VL (Qwen Team, 2025), GLM-4.1V (Hong et al., 2025),

and Gemma-3 (Kamath et al., 2025). Since OpenAI has not publicly disclosed GPT-5.2's vision encoder specifications, we apply the same configuration as Qwen3-VL based on common ViT conventions.

*Table 7.* Patch alignment configurations by model.

| Model | Patch Size | Image Size | Notes |
|---|---|---|---|
| Qwen3-VL | $16 \times 16$ | $1024 \times 1024$ | Dynamic resolution |
| GLM-4.1V | $14 \times 14$ | $1008 \times 1008$ | Dynamic resolution |
| Gemma-3 | $14 \times 14$ | $896 \times 896$ | Fixed resize |
| GPT-5.2 | $16 \times 16$ | $1024 \times 1024$ | Assumed (not disclosed) |

For models with dynamic resolution support (Qwen3-VL, GLM-4.1V), we select image sizes that are exact multiples of their respective patch sizes to ensure clean patch boundaries. Since OpenAI has not publicly disclosed GPT-5.2's vision encoder specifications, we apply the same configuration as Qwen3-VL based on common ViT conventions. To ensure all board positions share identical patch-relative offset, cell size is constrained to be an exact multiple of patch size.

**Visualization.** Figure 14 visualizes the four alignment conditions for both games. Red grid lines indicate patch boundaries ($16 \times 16$ pixels). Green dots mark aligned element centers (piece centers for chess, intersections for Gomoku) that fall at the target offset within each patch.

### B.4. Resolution Divisibility Test

Table 8 specifies the resolution configurations for each patch size group.

*Table 8.* Resolution configurations by patch size.

| Patch Size | Models | Divisible | Non-Divisible |
|---|---|---|---|
| $16 \times 16$ | Qwen3-VL, GPT-5.2 | 1024, 512, 384 | 1010, 510, 370 |
| $14 \times 14$ | GLM-4.1V, Gemma-3 | 1008, 896, 392 | 1010, 900, 400 |

Divisible resolutions correspond to exact multiples of patch size (e.g., $1024 = 64 \times 16$, $1008 = 72 \times 14$). All boards use medium density (chess: 16–20 pieces; Gomoku: 40–50% occupancy) with layout parameters scaled proportionally to maintain consistent visual appearance.

### B.5. Visual Richness Test

**2D Flat Style** Minimalist rendering using solid colors and simple geometric shapes: uniform board colors (light: RGB 240, 217, 181; dark: RGB 181, 136, 99 for chess), flat piece icons, and simple circles for Gomoku stones.

**3D Rendered Style** Realistic rendering using PNG assets with textures and shading: wood grain board textures, gradient-shaded stones with specular highlights for Gomoku, and detailed 3D piece renderings with shadows for chess.

Figure 15 illustrates the two visual styles for both games.

### B.6. Structural Shift Analysis: Implementation Details

This section provides implementation details for the structural shift metrics defined in Section 3.

#### B.6.1. PIECE MATCHING VIA HUNGARIAN ALGORITHM

We match predicted pieces to ground-truth pieces using the Hungarian algorithm (Kuhn, 1955) with Chebyshev distance. Given predicted board $\hat{B}$ and ground-truth board $B$, we construct a cost matrix $C$ where:

$$C_{ij} = \begin{cases} d_\infty(p_i, \hat{p}_j) & \text{if } B[p_i] = \hat{B}[\hat{p}_j] \text{ and } d_\infty \leq 1 \\ \infty & \text{otherwise} \end{cases} \quad (4)$$

where $d_\infty(a, b) = \max(|a_r - b_r|, |a_c - b_c|)$ is the Chebyshev distance. The constraint $d_\infty \leq 1$ restricts matches to off-by-one errors. For each matched pair, we record the shift direction $d(p) = (\hat{p}_r - p_r, \hat{p}_c - p_c) \in D$, where $D$ is the set of eight possible directions.

#### B.6.2. MONTE CARLO BASELINE CONSTRUCTION

We construct a Monte Carlo baseline ($K = 200$ simulations) by randomly permuting shift directions among piece positions while preserving the total count of each direction:

1. Let $\{d_1, d_2, \ldots, d_n\}$ be the observed shift directions

2. For each simulation $k = 1, \ldots, K$:

    (a) Randomly permute the direction assignments across positions

    (b) Compute $R_{\text{adj}}^{(k)}$ under the permuted assignment

3. Report baseline mean, standard deviation, and 95th percentile

Ratios substantially exceeding this baseline indicate that spatial adjacency increases the probability of same-direction shifts—evidence that localized regions are perceptually displaced as a unit.

#### B.6.3. STATISTICAL TESTING

For individual boards, we test significance using a one-sided binomial test:

$$H_0 : R_{\text{adj}} = 1/8 \quad \text{vs.} \quad H_1 : R_{\text{adj}} > 1/8 \quad (5)$$

We report the percentage of boards achieving $p < 0.05$.

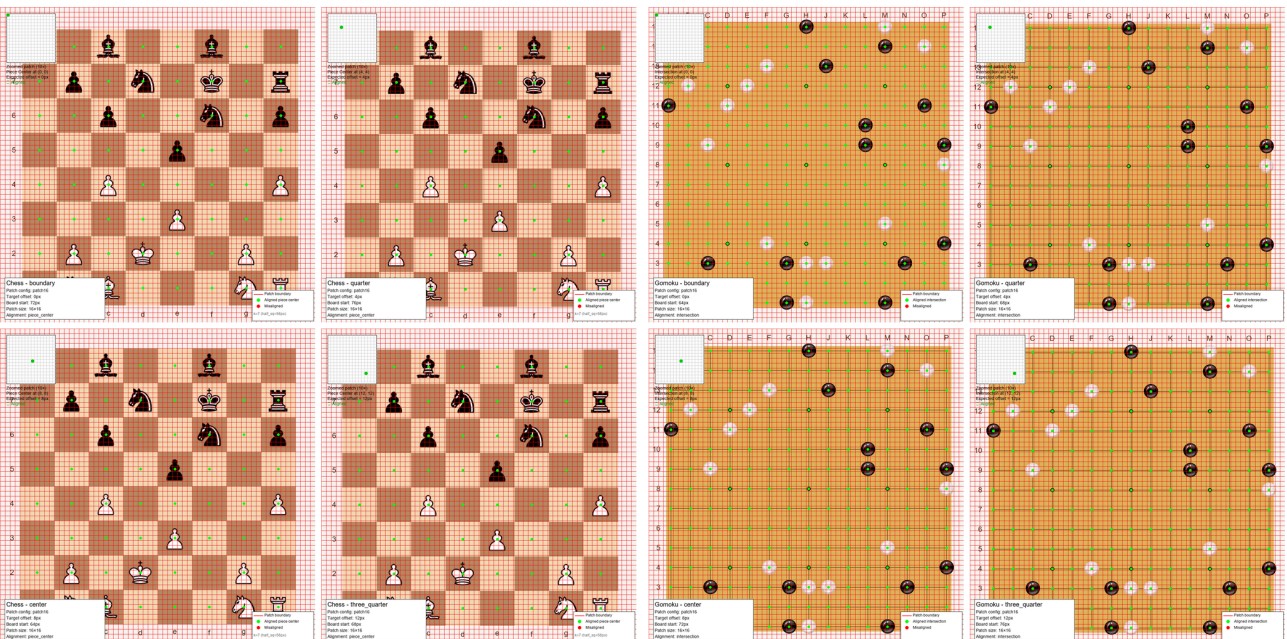

*Figure 14.* **Patch alignment visualization.** Four alignment conditions (boundary, quarter, center, three-quarter) for Gomoku (right) and Chess (left), showing how board elements align relative to $16 \times 16$ patch boundaries (red grid). Green dots indicate aligned centers at the target offset—intersection points for Gomoku, piece centers for Chess.

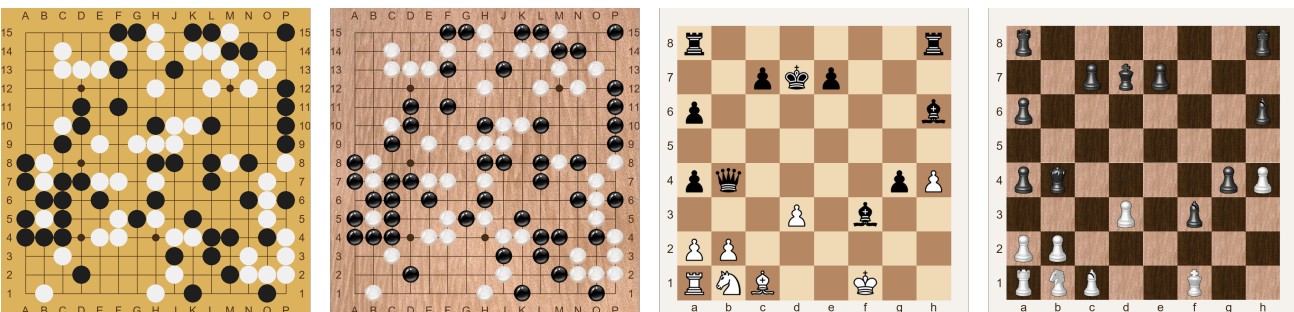

*Figure 15.* **Visual richness comparison.** Gomoku (left two) and Chess (right two), each showing 2D flat style versus 3D rendered style. 2D uses solid colors and simple geometric shapes; 3D uses wood grain textures, gradient-shaded stones, and detailed piece renderings with shadows.

### B.6.4. SUPPORTING METRICS

**Moran's I Spatial Autocorrelation**  We compute Moran's I separately for row and column shift components:

$$I = \frac{n}{\sum_{i,j} w_{ij}} \cdot \frac{\sum_{i,j} w_{ij}(z_i - \bar{z})(z_j - \bar{z})}{\sum_i (z_i - \bar{z})^2} \quad (6)$$

where $w_{ij} = 1$ if $d_\infty(p_i, p_j) \leq 3$, and $z_i$ is the shift component. Positive values indicate spatial clustering of similar shifts.

**Largest Connected Component**  Using 8-connectivity, we compute the size of the largest connected component among pieces sharing the dominant shift direction. High ratios suggest block-like displacement.

### B.6.5. TWO-LEVEL WEIGHTED AVERAGING

To ensure balanced representation across tests with varying sample sizes:

1. **Level 1**: Compute metrics within each test category (density, patch, resolution, richness)

2. **Level 2**: Average across categories with equal weights

# C. Rule-Following Test Details

## C.1. Diagnostic Matrix Specifications

### C.1.1. SINGLE-STATE TESTS

**Chess Rule-Free**  Five categories: *Same File* (vertical alignment), *Same Rank* (horizontal alignment), *Diagonal* (diagonal relationship), *Direction* (8 cardinal/ordinal directions), and *Path Clear* (orthogonal path obstruction).

**Chess Rule-Based**  Covers all six piece types plus castling:

- *King*: One-square movement (subtypes: valid, too_far).

- *Queen*: Diagonal/straight with path blocking (subtypes: clear_path, blocked_path, invalid_move).

- *Rook*: Straight-line with path blocking (subtypes: clear_path, blocked_path, not_straight).

- *Bishop*: Diagonal with path blocking (subtypes: clear_path, blocked_path, not_diagonal).

- *Knight*: L-shape pattern (subtypes: valid_move, invalid_move).

- *Pawn*: Forward movement with starting rules (subtypes: valid_forward, backward, sideways, diagonal_no_capture, too_far).

- *Castling*: Through-check and in-check conditions.

**Xiangqi Rule-Free**  Six categories: *Same File*, *Same Rank*, *Diagonal*, *Direction*, *Path Clear* (same as Chess), plus *River Position* (whether square is north of river).

**Xiangqi Rule-Based**  Covers seven piece types with Xiangqi-specific constraints:

- *Rook*: Orthogonal movement with path blocking.

- *Cannon*: Orthogonal non-capture moves.

- *Knight*: L-shape with leg blocking.

- *Bishop*: Diagonal 2-step with eye blocking; cannot cross river.

- *Advisor*: Diagonal within $3 \times 3$ palace.

- *King*: Orthogonal within palace.

- *Pawn*: Forward only before river; sideways after crossing.

### C.1.2. MULTI-STATE TESTS

Each state is rendered as a separate image with "State N" label. Images are presented chronologically with explicit mapping in prompt.

Figure 16 illustrates the design principle behind multi-state tests: we construct sequences with identical final positions but different histories that yield opposite legality judgments, ensuring models cannot solve tasks by examining only the last frame.

**Chess Rule-Free**  Four categories: *Movement Detection* (piece moved between squares), *Sequence Order* (movement order across states), *State Comparison* (piece returned to start), *Position Tracking* (piece ever at square).

**Chess Rule-Based**  History-dependent rules:

- *En Passant*: Legality based on opponent pawn's previous move.

- *Castling*: Legality based on piece movement history.

- *Event Recognition*: Multiple-choice identification of en passant/castling from sequences.

- *Direct Movement*: Whether piece can move directly between positions.

**Xiangqi Rule-Free**  Same four categories as Chess, adapted to Xiangqi.

**Xiangqi Rule-Based**  Move legality requiring state context:

- *Move Legality*: Knight blocking, bishop eye blocking, cannon capture validation, flying general rule, rook path blocking.

- *Perpetual Check*: Whether move violates perpetual check rule (6-state sequences).

## C.2. Rule Complexity Ladder Specifications

Each level is implemented in two modes: *Predictive* (shows N states, asks if a move is possible) and *Explicit* (shows N+1 states including the result, asks if the completed move is legal).

### C.2.1. LEVELS 1–2: PATTERN RECOGNITION

**Level 1: Basic Movement Pattern Recognition**  Level 1 tests whether models can recognize basic movement patterns for each piece type without path blocking or special rules.

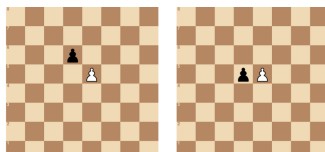 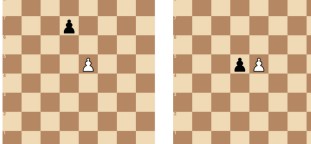 

Type 1: Sequential verification with invalid transition

Question: Can white pawn do en passant?

Answer: Yes/No/Unknown

Type 2: Minimal-change discrimination with valid transition

Question: Can white pawn do en passant?

Answer: Yes/No/Unknown

Type 3: A complete state sequence of en passant

Question: What happened?

Answer: Normal capture/En passant/Castling/ None of Above/Unsure

*Figure 16.* **Enforcing multi-state integration:** sequences with identical final positions but different histories yield opposite legality judgments.

**Chess L1    Pieces Tested:** Knight, Bishop, Rook, Queen, King, and Pawn.

**Test Structure:**

- **Predictive Mode**: Single state showing piece position; question asks "Can the [Piece] at [start] move to [target]?"

- **Explicit Mode**: Two states showing piece before and after movement; question asks "Is this a legal move according to chess rules?"

**Valid Moves:**

- *Knight*: L-shape moves $(\pm2, \pm1)$ or $(\pm1, \pm2)$

- *Bishop*: Diagonal moves in any diagonal direction

- *Rook*: Horizontal or vertical moves

- *Queen*: Straight or diagonal moves

- *King*: One square in any of 8 directions

- *Pawn*: Forward 1 square (or 2 from starting rank)

**Invalid Moves:**

- *Knight*: Straight or diagonal moves

- *Bishop*: Horizontal or vertical moves

- *Rook*: Diagonal moves

- *Queen*: L-shape moves

- *King*: Moves of distance $\geq 2$

- *Pawn*: Backward, sideways, or distance $\geq 3$ moves

**Xiangqi L1**    Level 1 tests basic orthogonal movement for pieces without blocking constraints.

**Pieces Tested:** Rook and Cannon (movement only, not capture).

**Valid Moves:** Orthogonal movement (horizontal or vertical) of any distance.

**Invalid Moves:** Diagonal movement of any distance.

**Level 2: Conditional Movement Rules**

**Chess L2: En Passant**    Level 2 tests en passant capture, which requires verifying multiple conditions: (1) the captured pawn started from its initial rank, (2) it made a double-step move, and (3) the capturing pawn is adjacent.

**State Sequence:**

- **Predictive Mode (3 states)**: State 1: Initial positions; State 2: White knight moves; State 3: Black pawn advances

- **Explicit Mode (4 states)**: Adds State 4 showing en passant capture result

**Valid Subtypes:**

- *all_conditions_met*: Black pawn double-steps from rank 7, white pawn adjacent

- *correct_pawn_identified*: Multiple pawns present, question correctly identifies the double-stepped pawn

**Invalid Subtypes:**

- *not_from_start*: Black pawn starts from rank 6 instead of rank 7

- *moved_one_square*: Black pawn advances only one square

- *not_adjacent*: White pawn not on adjacent file

- *multi_pawn_confusion*: Two black pawns present; asked pawn only moved one square

- *wrong_pawn_asked*: Pawn A double-stepped, but question asks about pawn B

**Xiangqi L2: Complex Movement Rules with Blocking**
Level 2 tests pieces with spatial constraints: blocking positions, palace restrictions, and river constraints.

**Pieces Tested:** Knight, Bishop, King, and Advisor.

**Knight:**

- Movement: L-shape $(\pm 2, \pm 1)$ or $(\pm 1, \pm 2)$

- Constraint: "Leg" position must be empty

- Subtypes: *valid*, *blocked*, *invalid_pattern*

**Bishop:**

- Movement: Exactly 2 squares diagonally

- Constraints: "Eye" position must be empty; cannot cross river

- Subtypes: *valid*, *blocked*, *cross_river*, *invalid_pattern*

**King:**

- Movement: One square orthogonally

- Constraint: Must stay within $3 \times 3$ palace

- Subtypes: *valid*, *outside_palace*, *diagonal*, *too_far*

**Advisor:**

- Movement: One square diagonally

- Constraint: Can only occupy 5 specific positions within palace

- Subtypes: *valid*, *orthogonal*, *outside_palace*, *too_far*

C.2.2. LEVELS 3–4: PATH AND GLOBAL CONSTRAINTS

**Level 3: Path and Obstruction Detection**

**Chess L3** Level 3 tests whether models can track path blocking across temporal changes for sliding pieces.

**Pieces Tested:** Rook, Bishop, and Queen.

**State Sequence:**

- **Predictive Mode (2 states)**: State 1: Initial positions with potential blocker; State 2: Blocker moves

- **Explicit Mode (3 states)**: Adds State 3 showing capture result

**Valid Subtypes:**

- *path_cleared*: Knight blocker jumps away from path; capture becomes possible

**Invalid Subtypes:**

- *still_blocked*: Queen blocker moves along the path to a different position; still blocks

- *path_blocked*: Knight jumps into the path; capture becomes impossible

- *invalid_pattern*: Attacker attempts illegal movement pattern (e.g., rook moving diagonally)

**Xiangqi L3: Capture Rules with Blocking** Level 3 tests capture mechanics for pieces with special capture constraints.

**Pieces Tested:** Knight, Bishop, Cannon, and Pawn.

**Knight and Bishop Capture:** Same blocking rules as Level 2, with an enemy piece at target position.

- Subtypes: *capture_valid*, *capture_blocked*

**Cannon Capture:**

- Constraint: Requires exactly one "screen" piece between cannon and target

- Subtypes: *capture_valid* (one screen), *no_screen*, *two_screens*

**Pawn Capture:**

- Before crossing river: Forward capture only

- After crossing river: Forward or sideways capture allowed

- King placement indicates own side for river crossing determination

- Subtypes: *forward_after* (valid), *sideways_after* (valid), *forward_before* (valid), *sideways_before* (invalid), *backward* (invalid), *diagonal* (invalid)

**Test Structure:**

- **Predictive Mode**: Single state with attacker and target; question asks "Can the [Piece] capture the piece at [target]?"

- **Explicit Mode**: Two states showing before and after capture

**Level 4: Global Legality Constraints**  Level 4 tests global constraints that depend on the entire board state, not just local piece positions.

**Chess L4: En Passant with Constraints**  En passant capture requires verifying multiple conditions simultaneously: timing (immediately after opponent's double-step), positioning (adjacent file), and legality (does not expose own King to check).

**State Sequence:**

- **Predictive Mode (2–3 states)**: Shows pawn positions before and after double-step; for timing violations, includes an intervening move.

- **Explicit Mode (3–4 states)**: Adds final state showing the en passant capture result.

**Valid Subtypes:**

- *valid*: All conditions met—opponent pawn double-stepped from starting rank, capturing pawn adjacent, no check exposure.

**Invalid Subtypes:**

- *missed_timing*: White moved another piece after Black's double-step; en passant window closed.

- *causes_check_pin*: Capturing pawn is pinned along file or rank; en passant would expose King to Rook/Queen check.

- *already_in_check*: King is currently in check from a distant piece; en passant does not resolve the check.

**Xiangqi L4: Flying General and Pawn Constraints** Level 4 tests two Xiangqi-specific global constraints: the Flying General rule (Kings cannot face each other on an open file) and pawn movement restrictions based on river crossing.

**Flying General Rule:** The two Kings cannot directly face each other on the same file with no pieces between them. A piece blocking this line cannot move away unless another blocker remains.

**State Sequence:**

- **Predictive Mode (1 state)**: Shows current board with Kings on same file and potential blocker.

- **Explicit Mode (2 states)**: Shows before and after the proposed move.

**Flying General Subtypes:**

- *single_blocker_leaves* (invalid): Only one piece blocks the Kings; moving it away causes Flying General.

- *blocker_stays_on_column* (valid): Blocker moves along the same file; still blocks Kings.

- *multiple_blockers* (valid): One blocker moves away but another remains.

- *kings_different_columns* (valid): Kings not on same file; piece moves freely.

**Pawn River Crossing:** Pawns gain sideways movement ability only after crossing the river (reaching opponent's half of the board).

**Pawn Subtypes:**

- *crossed_river_sideways* (valid): Pawn has crossed river; sideways move allowed.

- *not_crossed_river_sideways* (invalid): Pawn before river; sideways move illegal.

C.2.3. LEVELS 5–6: MULTI-CONDITION AND
     TEMPORAL REASONING

**Level 5: Multi-Condition Integration**  Level 5 requires simultaneous verification of 4–5 conditions, combining temporal history with spatial constraints.

**Chess L5: Castling with History and Two Check Rules**
Castling legality requires: (1) King has never moved, (2) Rook has never moved, (3) path is clear, and (4–5) King does not start in, pass through, or end in check. Level 5 tests conditions (1)–(2) plus two of the three check rules.

**Check Rule Combinations Tested:**

- In-check + Through-check

- In-check + Into-check

- Through-check + Into-check

**State Sequence:**

- **Predictive Mode (2–3 states)**: Shows movement history; question asks if castling is possible.

- **Explicit Mode (3–4 states)**: Adds final state showing castling result.

**Valid Subtypes:**

- *valid_other_moved*: Other pieces moved; King and Rook remained stationary; tested check rules satisfied.

**Invalid Subtypes:**

- *invalid_king_moved*: King moved to temporary square and returned; castling rights lost.

- *invalid_rook_moved*: Rook moved to temporary square and returned; castling rights lost.

- *invalid_check_violation*: Enemy piece attacks one or both tested critical squares.

**Xiangqi L5: Capture under Pin and Flying General** Tests whether a piece can legally capture, considering absolute pins (piece blocks attack on own King) and Flying General constraint (piece is sole blocker between Kings).

**State Sequence:**

- **Predictive Mode (1 state)**: Shows board with potential capturer, target, and constraint pieces.

- **Explicit Mode (2 states)**: Shows before and after capture.

**Invalid Subtypes:**

- *pinned_by_rook*: Piece pinned by enemy Rook along file/rank; sideways capture exposes King.

- *pinned_by_cannon*: Piece pinned by enemy Cannon (with screen); sideways capture exposes King.

- *flying_general_capture*: Piece is only blocker between Kings; capture causes Flying General.

**Valid Subtypes:**

- *not_pinned*: Capturer not on line between King and attacker; captures freely.

- *multiple_blockers*: Another piece remains to block Kings after capture.

- *capture_along_pin*: Pinned Rook captures the pinning piece along the pin line.

**Cannon Capture Handling:** When the capturing piece is a Cannon, an additional screen piece is placed between the Cannon and target to enable the capture, following Xiangqi's Cannon capture rule.

**Level 6: Advanced Temporal Reasoning** Level 6 requires tracking 5–6 conditions across extended state sequences, testing the limits of temporal reasoning.

**Chess L6: Castling with Full Constraints** Extends Level 5 to test all three check rules simultaneously: King cannot be in check, pass through check, or end in check.

**State Sequence:**

- **Predictive Mode (2–3 states)**: Shows movement history and potential attackers.

- **Explicit Mode (3–4 states)**: Adds final state showing castling result.

**Subtypes:** Same as Level 5, but check violations now test all three rules (in/through/into) rather than two.

**Xiangqi L6: Perpetual Check and Perpetual Chase** Tests detection of illegal repetitive patterns. In Xiangqi, perpetual check (continuously checking the King) and perpetual chase (continuously threatening a piece) are illegal after three consecutive occurrences.

**Rule Explanation:** A note is included in each question: "In Xiangqi, Perpetual Check and Perpetual Chase are illegal after 3 consecutive occurrences."

**State Sequence (6–7 states):**

- **Predictive Mode (6 states)**: Shows three cycles of check/chase pattern; asks if fourth is legal.

- **Explicit Mode (7 states)**: Adds final state showing the fourth check/chase move.

**Perpetual Check Subtypes:**

- *perpetual_check_violation* (invalid): Rook checks King three times in repeating pattern; fourth check illegal.

- *pattern_broken* (valid): Check pattern interrupted by non-checking move; counter resets.

**Perpetual Chase Subtypes:**

- *perpetual_chase_violation* (invalid): Rook chases Cannon three times; fourth chase illegal.

**Chase Pattern Structure:** Each chase cycle consists of: (1) Rook threatens Cannon on same rank/file, (2) Cannon escapes to different rank and file, (3) Rook moves legally to threaten again. The generator ensures all Rook moves follow legal straight-line movement with clear paths.

*Table 9.* Chess density test with randomized placement: piece perception accuracy (%).

| Model | Low | Medium | High |
|---|---|---|---|
| Gemma-3 27B | 9.92 | 11.51 | 13.46 |
| GPT-5.2 | 99.33 | 99.80 | 99.77 |
| Qwen3-VL 8B | 80.70 | 71.42 | 72.58 |
| Qwen3-VL 30B | 85.76 | 83.67 | 87.78 |
| Qwen3-VL 235B | 96.21 | 95.42 | 95.91 |

*Table 10.* Patch alignment test: Piece perception accuracy (%) across conditions.

| Model | Chess | | | | Gomoku | | | |
|---|---|---|---|---|---|---|---|---|
| | Boundary | Quarter | Center | 3/4 | Boundary | Quarter | Center | 3/4 |
| Gemma-3 27B | 36.41 | 36.63 | 33.52 | 36.53 | 4.38 | 4.08 | 4.03 | 5.40 |
| GLM-4.1V 9B | 66.30 | 70.31 | 72.46 | 71.62 | 22.80 | 22.01 | 21.80 | 24.69 |
| GPT-5.2 | **98.34** | **98.34** | 96.49 | 96.91 | 41.42 | 43.79 | 49.84 | 45.01 |
| Qwen3-VL 8B | 86.13 | 84.78 | 84.63 | 87.59 | 39.94 | 39.89 | 41.14 | 39.28 |
| Qwen3-VL 30B | 85.39 | 87.74 | 88.40 | 90.85 | 48.74 | 45.65 | 47.04 | 45.38 |
| Qwen3-VL 235B | 96.79 | 95.88 | **96.88** | **96.97** | 70.08 | 69.25 | 68.29 | 65.67 |

### C.2.4. MATCHED-SAMPLE ANALYSIS FOR SIMULATION GAP

This section provides aggregation details for the matched-sample analysis defined in Section 4. The intersection set $\mathcal{S}_{\text{valid}}$ is constructed as specified in Equation (3).

**Aggregation Procedure**   For each (model, level) pair, we first compute Explicit and Predictive accuracies separately for Chess and Xiangqi on their respective intersection sets, then average across both games. The aggregated curves in Figure 9 are produced by further averaging across all models. This two-level averaging ensures equal weighting across games and models regardless of intersection set sizes.

## D. Extended Results

This section presents detailed results not included in the main text due to space constraints.

### D.1. Perception Test Results

#### D.1.1. DENSITY TEST

**Chess Density Confound Control**   Table 9 presents chess perception accuracy using randomized piece placements, controlling for the positional canonicity confound in the simulated-gameplay data (Figure 3). Under controlled placement, the density-accuracy trend is no longer consistently present across models.

#### D.1.2. PATCH ALIGNMENT TEST

**Detailed Results**   Table 10 presents piece perception accuracy across four alignment conditions.

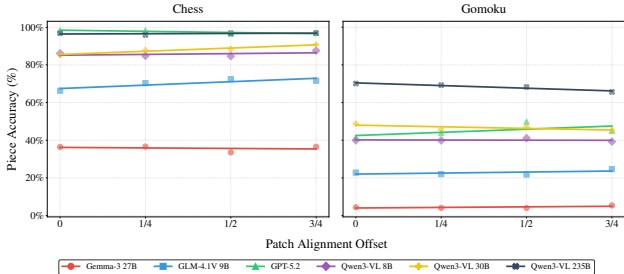

*Figure 17.* **Piece accuracy across patch alignment offsets.** Horizontal axis: offset of piece centers relative to patch boundaries—0 (boundary), 1/4 (quarter), 1/2 (center), 3/4 (three-quarter). Flat trends indicate patch alignment has minimal effect on perception accuracy.

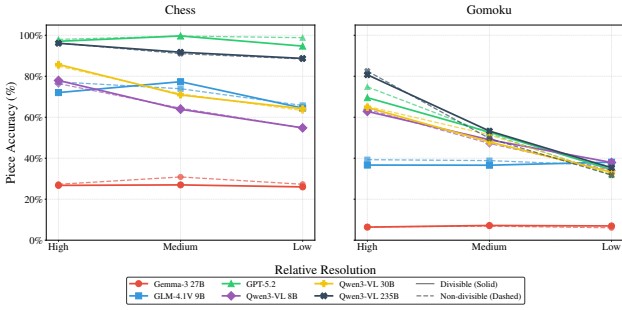

*Figure 18.* **Resolution trends: divisible (solid) vs. non-divisible (dashed).** Since models use different patch sizes, we normalize to relative resolution levels (High/Medium/Low) rather than absolute pixel counts. Overlapping solid and dashed lines indicate divisibility has minimal impact; downward slopes show resolution magnitude affects accuracy.

**Trend Analysis**   Figure 17 visualizes accuracy trends across alignment offsets. The relatively flat slopes confirm that patch alignment has minimal effect on perception accuracy.

#### D.1.3. RESOLUTION DIVISIBILITY TEST

**Detailed Results**   Table 11 presents piece perception accuracy across resolution conditions.

**Trend Analysis**   Figure 18 visualizes the trends. Overlapping solid (divisible) and dashed (non-divisible) lines confirm that divisibility has minimal impact, while downward slopes show that resolution magnitude substantially affects accuracy. Gemma-3 27B is an exception, showing flat performance across resolutions, likely because its accuracy is already near floor level.

#### D.1.4. VISUAL RICHNESS TEST

**Detailed Results**   Table 12 presents piece perception accuracy for 2D and 3D rendering styles.

*Table 11.* Resolution test: Piece perception accuracy (%) across conditions.

| Model | Chess | | | | | | Gomoku | | | | | |
| | Divisible | | | Non-Divisible | | | Divisible | | | Non-Divisible | | |
| | High | Med | Low | High | Med | Low | High | Med | Low | High | Med | Low |
|---|---|---|---|---|---|---|---|---|---|---|---|---|
| Gemma-3 27B | 26.75 | 26.99 | 26.01 | 27.29 | 30.86 | 27.30 | 6.38 | 7.18 | 6.95 | 6.52 | 6.86 | 6.07 |
| GLM-4.1V 9B | 72.04 | 77.34 | 64.47 | 77.24 | 73.92 | 65.75 | 36.66 | 36.59 | **38.06** | 39.30 | 38.84 | **36.48** |
| GPT-5.2 | **97.07** | **99.67** | **94.72** | **98.00** | **99.61** | **98.84** | 69.56 | 52.75 | 34.41 | 74.84 | **52.07** | 31.79 |
| Qwen3-VL 8B | 77.95 | 63.88 | 54.82 | 76.44 | 64.43 | 54.79 | 62.85 | 49.01 | 37.92 | 63.90 | 47.21 | 34.01 |
| Qwen3-VL 30B | 85.78 | 70.89 | 64.16 | 84.99 | 71.41 | 63.26 | 64.80 | 47.97 | 33.23 | 65.31 | 51.43 | 32.18 |
| Qwen3-VL 235B | 96.10 | 91.74 | 88.70 | 96.20 | 91.04 | 88.46 | **80.74** | **53.27** | 35.33 | **82.54** | 49.65 | 31.77 |

*Table 12.* Visual Richness test: Piece perception accuracy (%) across conditions.

| Model | Chess | | Gomoku | |
| | 2D Flat | 3D Rendered | 2D Flat | 3D Rendered |
|---|---|---|---|---|
| Gemma-3 27B | 26.16 | 25.32 | 5.48 | 6.63 |
| GLM-4.1V 9B | 74.85 | 75.97 | 39.31 | 37.44 |
| GPT-5.2 | **96.59** | **99.01** | 72.37 | 68.69 |
| Qwen3-VL 8B | 77.54 | 81.19 | 61.66 | 60.90 |
| Qwen3-VL 30B | 86.42 | 90.35 | 67.96 | 62.73 |
| Qwen3-VL 235B | 96.33 | 97.97 | **77.11** | **68.86** |

*Table 13.* Rule Complexity Ladder Verification Rate (%)

| Model | Level 1 | | Level 2 | | Level 3 | | Level 4 | | Level 5 | | Level 6 | |
| | Exp. | Pred. | Exp. | Pred. | Exp. | Pred. | Exp. | Pred. | Exp. | Pred. | Exp. | Pred. |
|---|---|---|---|---|---|---|---|---|---|---|---|---|
| Gemma-3 27B | 44.0 | 89.0 | 12.0 | 32.0 | 3.0 | 26.0 | 2.0 | 12.0 | 1.0 | 10.0 | 0.0 | 1.0 |
| GLM-4.1V 9B | 30.0 | 99.0 | 10.0 | 52.0 | 13.0 | 35.0 | 9.0 | 12.1 | 10.0 | 8.0 | 3.0 | 1.0 |
| GPT-5.2 | 97.0 | 99.0 | 98.0 | 100.0 | 99.5 | 99.5 | 99.5 | 99.5 | 99.0 | 98.0 | 64.0 | 77.5 |
| Qwen3-VL 235B | 87.0 | 97.0 | 88.0 | 96.0 | 93.0 | 98.0 | 79.8 | 89.8 | 90.0 | 92.0 | 77.0 | 77.0 |
| Qwen3-VL 30B | 68.0 | 96.0 | 58.0 | 89.0 | 83.0 | 85.0 | 63.8 | 70.7 | 63.0 | 73.0 | 47.0 | 62.0 |
| Qwen3-VL 8B | 52.0 | 100.0 | 53.0 | 88.0 | 82.0 | 84.0 | 53.6 | 81.8 | 76.0 | 71.0 | 42.0 | 42.0 |

**Trend Analysis**  Figure 19 visualizes the comparison. On chess, most models perform slightly better with 3D style (1–4 percentage point gains), with GPT-5.2 improving from 96.6% to 99.0%. On Gomoku, the pattern reverses: Qwen3-VL 235B drops from 77.1% to 68.9%, and Qwen3-VL 30B shows a similar 5-point decline.

## D.2. Rule-Following Test Results

### D.2.1. COMPLEXITY LADDER

**Verification Rates**  Table 13 presents the verification rates across complexity levels for all models.

**Perception-Reasoning Dissociation Examples**  Beyond the en passant example in the main text, castling legality tests reveal similar dissociation patterns. Qwen3-VL 235B correctly verifies during the perception phase that "White's bishop is on a5," yet concludes that Black's queenside castling is legal, despite this bishop controlling the d8 square through which the black king must pass. The logical chain "bishop on a5 → attacks d8 → king passes through d8 → illegal" is never triggered. In both cases, visual sym-bols are correctly encoded but fail to transform into logical constraints.

**Per-model Simulation Gap**  Figure 20 presents the simulation gap for individual models. The gap emerges consistently across all evaluated models, though with varying severity. At Level 6, Qwen3-VL 8B exhibits the largest gap (43 points: 75% vs 32%), while GPT-5.2 shows the smallest (8 points: 88% vs 80%). Scaling within the Qwen3-VL family does not eliminate this limitation: the 235B variant still exhibits a 39-point gap at Level 6 (80% vs 41%), comparable to the 8B model. Scaling from 8B to 235B does not close this gap, pointing to a systematic architectural limitation rather than a capacity one.

### D.2.2. ABLATION STUDY DETAILS

**Text Only**  Figure 21 presents the text-only and image-based (verified) accuracy separately for Chess and Xiangqi on matched samples (L3–L6). The dual-bottleneck pattern described in Section 4.2.5 holds consistently across both games.

**CoT Scratchpad Prompt**  The following scratchpad instruction is appended to the original prompt immediately after the main question:

> Analyze this step-by-step using the following scratchpad format:
> (1) State Identification: List the exact start and end co-ordinates of the moving piece.
> (2) Rule Retrieval: State the exact movement and capture rules for this specific piece type.
> (3) Trajectory Mapping: List every square the piece will pass through.
> (4) Constraint Checking: Check the start, path, and end squares against all rules and constraints identified in Step 2.
> (5) Final Conclusion: Based on the above, {question} (Yes/No/Unknown)

where {question} is replaced with the original test question for each case.

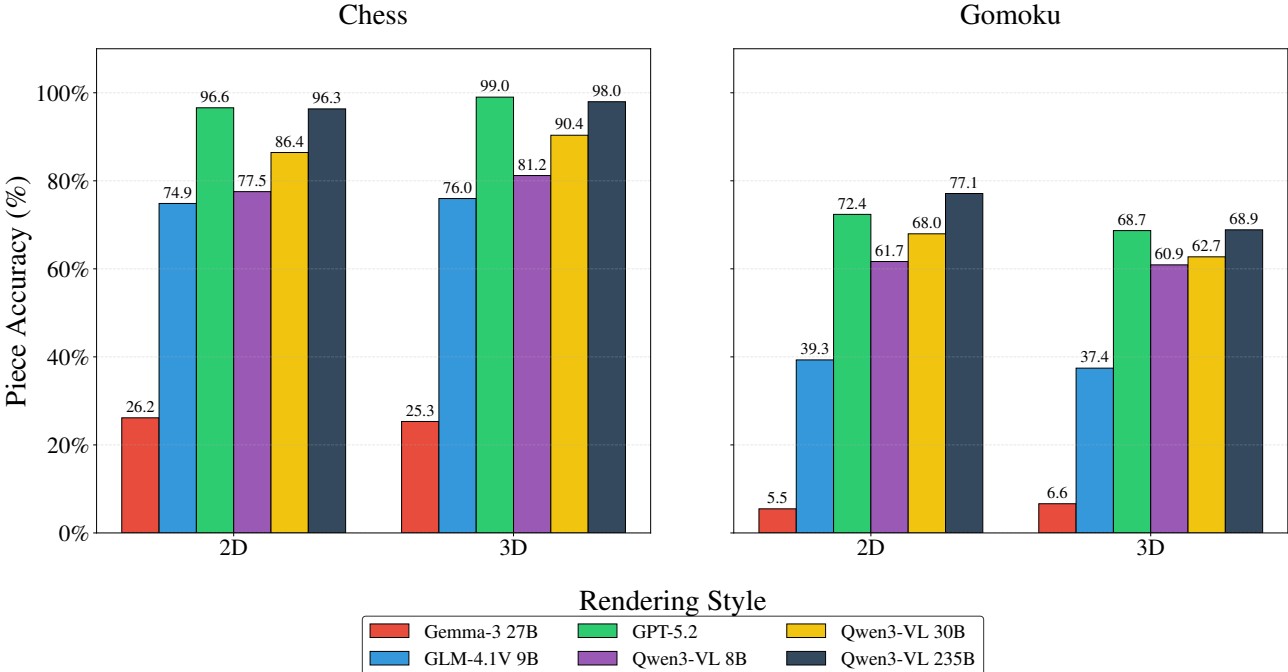

*Figure 19.* **Visual richness impact: 2D vs. 3D.** Accuracy differences are small and inconsistent across games, indicating visual complexity has limited systematic effect.

**CoT Ablation Per-Game Breakdown** Figure 22 presents the CoT and original accuracy separately for Chess and Xiangqi on matched samples (L3–L6). The patterns described in Section 4.2.5 hold consistently across both games.

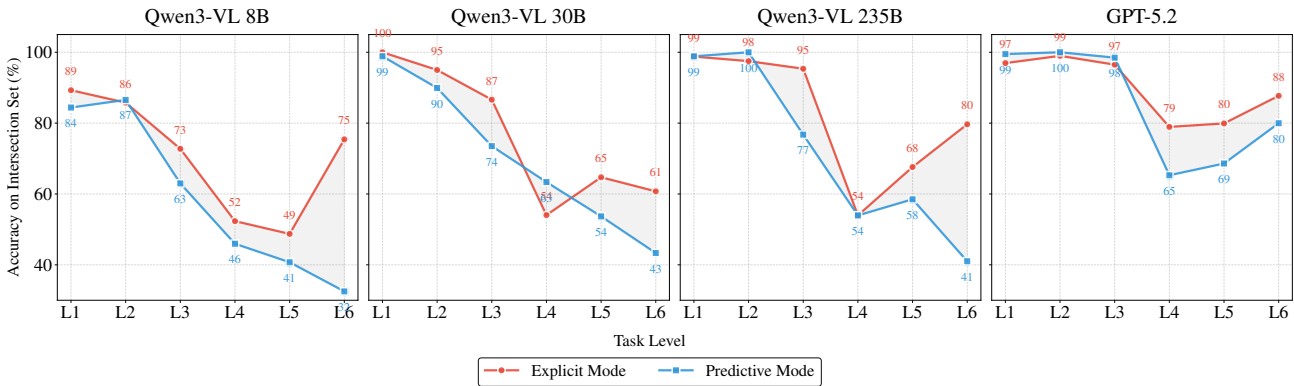

*Figure 20.* **Per-model simulation gap.** The gap between Explicit and Predictive modes persists across all models and scales. Scaling from 8B to 235B does not eliminate the limitation; even GPT-5.2 exhibits consistent gaps at higher complexity levels.

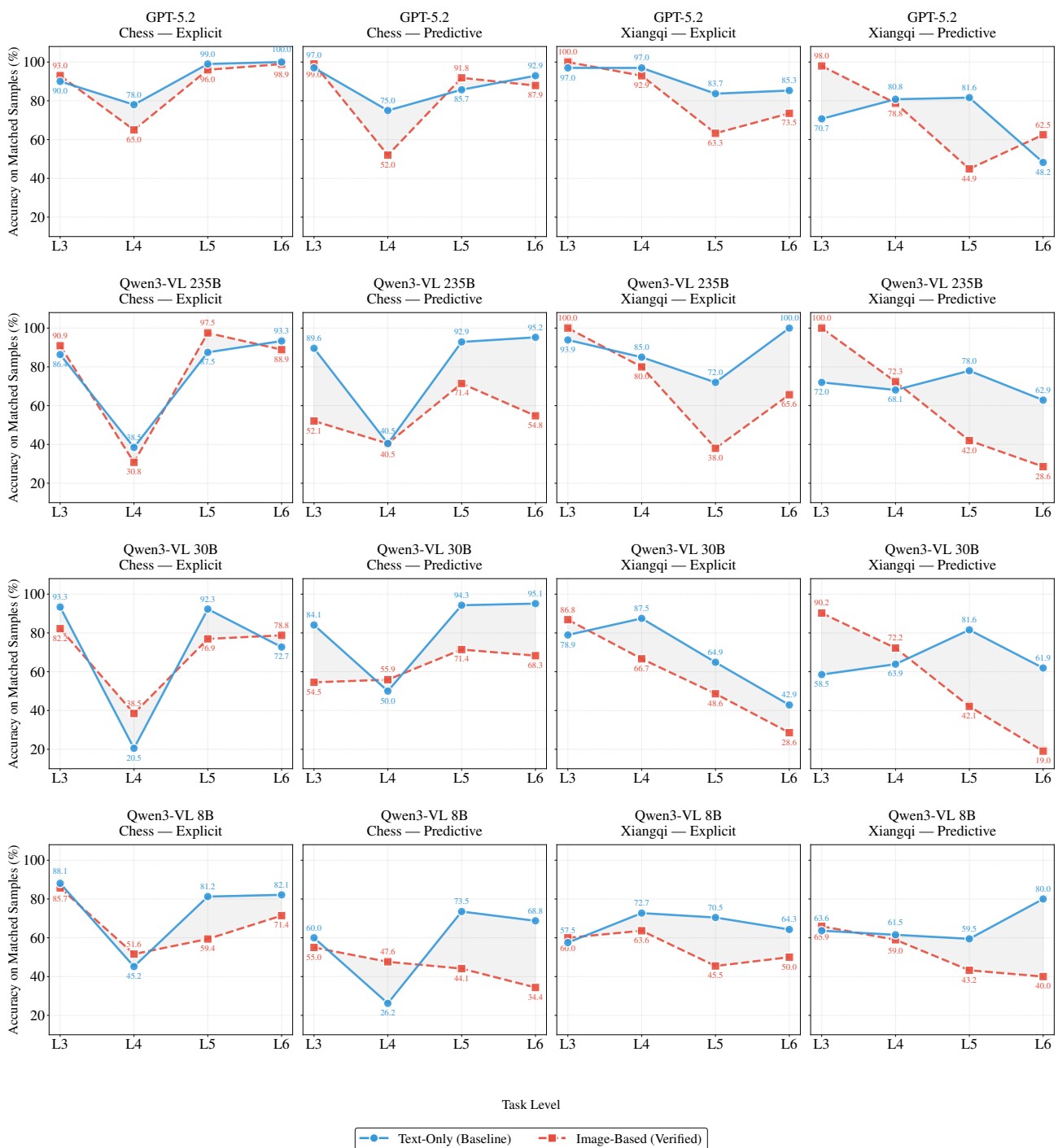

*Figure 21.* Text-only (FEN) vs. image-based (verified) reasoning accuracy by game on matched samples (L3–L6).

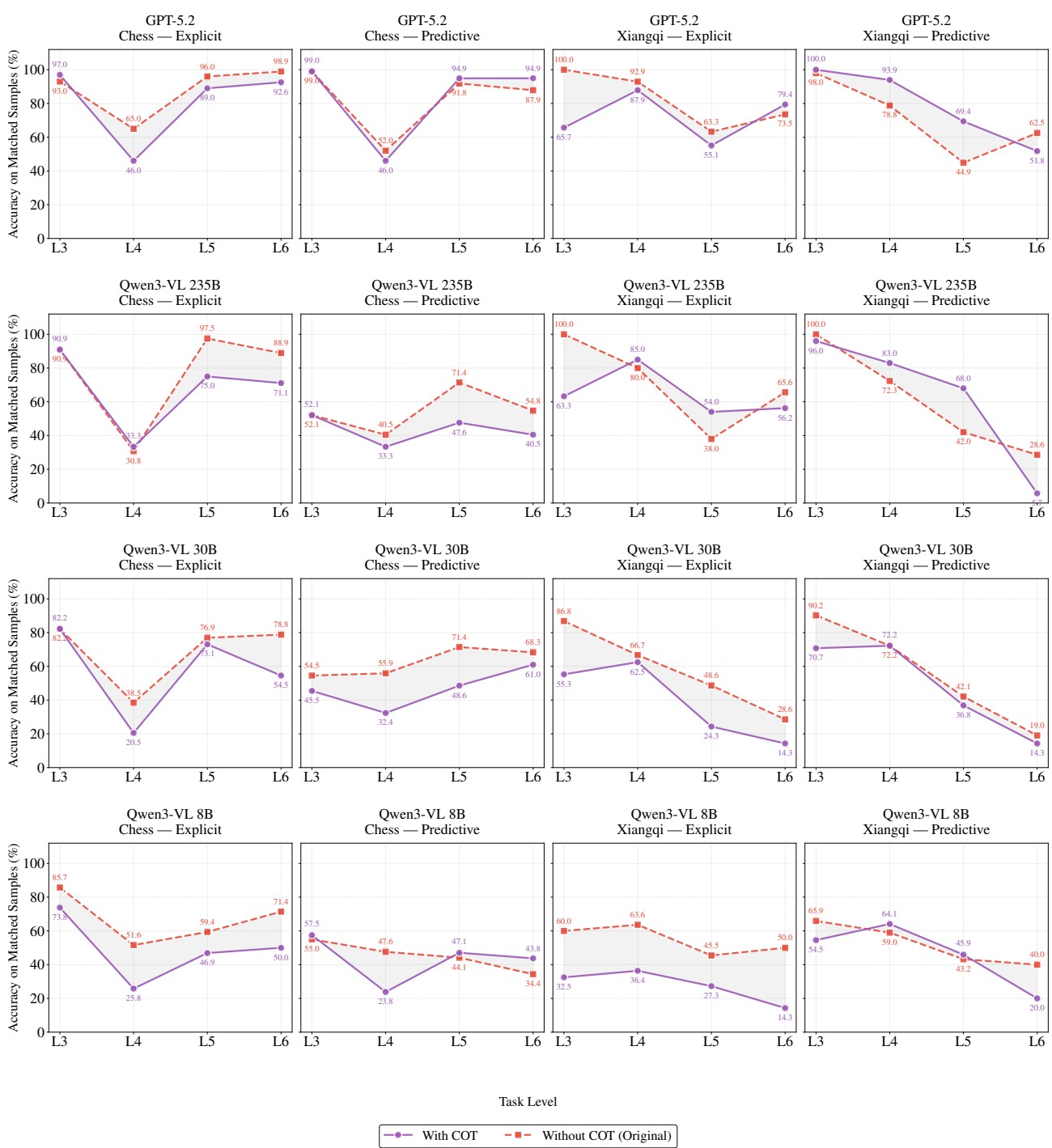

*Figure 22.* CoT ablation vs. original prompting by game on matched samples (L3–L6).

