# OpenReview forum: "Seeing Without Understanding: Disentangling Perception, Reasoning, and Simulation in VLM Gameplay"
_ICML.cc/2026/Conference — ICML 2026 regular_

### Official Review · Reviewer_1cy3 · 2026-03-09

**Soundness:** 3
**Presentation:** 3
**Significance:** 3
**Originality:** 3
**Overall Recommendation:** 3
**Confidence:** 2

**Summary:**

This paper introduces a thoughtfully designed, two-stage diagnostic framework to evaluate the perception, rule-following, and internal simulation capabilities of Large Vision-Language Models (VLMs) in grid-based board games (Chess, Gomoku, Xiangqi). To isolate reasoning failures from pure perception errors, the authors employ a verification-gating mechanism. The study uncovers three primary phenomena: (1) structurally coordinated, directionally biased off-by-one perception shifts; (2) a severe dissociation where rule-based reasoning collapses at higher complexities despite verified perception; and (3) a substantial "simulation gap" between retrospective verification and predictive reasoning.

Overall, the ambition to causally attribute VLM failure modes rather than merely reporting aggregate performance is highly commendable. The structural shift analysis utilizing Chebyshev-bounded Hungarian matching is particularly innovative. However, despite the elegant framework, the manuscript suffers from critical experimental blind spots. The absence of symbolic text-only baselines, an unaddressed data-generation confound in the density analysis, the omission of Chain-of-Thought ablations, and the exclusion of key contemporary diagnostic benchmarks significantly weaken the paper's sweeping claims regarding "fundamental architectural bottlenecks."

**Compliance With Llm Reviewing Policy:**

Affirmed.

**Final Justification:**

My final recommendation remains Weak Reject. I view this paper as thoughtful and technically interesting. The proposed two-stage diagnostic framework is well designed, the verification-gating setup is a principled way to reduce perception-reasoning confounds, and the structural analysis of coordinated perception errors is original and insightful. The paper is also clearly written, and I believe the benchmark construction and diagnostic perspective are valuable contributions.

The rebuttal improved my evaluation in several respects. In particular, I appreciate the added text-only baseline, the controlled random-placement experiment for the density analysis, the explicit CoT ablation, and the authors’ willingness to soften the overly strong “causal isolation” language. These additions address several important concerns from my original review and make the empirical story more careful and credible.

However, I still do not think the current paper fully supports its strongest conclusions. The most important remaining issue is one of framing and external validity rather than a missing ablation. The added text-only baseline is useful, but it also makes the interpretation more delicate: if a substantial part of the observed failure remains after replacing visual input with symbolic input, then the evidence for a specifically multimodal or architectural bottleneck is weaker than the original framing suggests. More broadly, I am not yet convinced that the paper’s decomposition into perception, rule reasoning, and simulation fully captures what should count as visual reasoning beyond the benchmark setting studied here.

Overall, I find the work promising, original, and useful as a diagnostic study, but I believe its current claims still overreach the evidence. The rebuttal strengthened the paper, but it did not change my final assessment, so I keep my original score unchanged. **But in accordance with my commitment during the discussion, I will reduce my negative confidence level to 2 for the AC to comprehensively consider the opinions of all reviewers. This will be at a borderline level, and our discussion has already reflected the contributions and deficiencies of this work, enabling the AC to make a comprehensive assessment. Therefore, I tend to take a neutral stance to minimize the impact on the final outcome of this article.**

**Key Questions For Authors:**

Q1: Can you provide a text-only control (e.g., FEN notation) for the rule-complexity ladder (Levels 3–6) using the same models? Does the reasoning accuracy still collapse when the visual modality is entirely removed?

Q2: How do you disentangle visual density from positional canonicity? Can you run a controlled perception experiment where density varies but the positional familiarity remains constant (e.g., placing random pieces rather than sampling from actual game trajectories)?

Q3: How much of the perception-reasoning dissociation (Figure 8) remains if you apply a matched-sample or difficulty-controlled analysis (similar to Equation 3), rather than simply conditioning on the per-level verified cases which may suffer from selection bias?

Q4: Did you experiment with explicit Chain-of-Thought prompts (e.g., requesting the model to explicitly list piece coordinates and legal trajectories before predicting the outcome)? If so, does the predictive simulation gap persist?

Based on the considerations outlined above and out of responsibility to the ICML review process, I assigned an initial score of 3: Weak reject. Should the authors effectively address these concerns in their rebuttal and demonstrate to me that their contributions are reliable and useful, I will consider raising the score.

**Limitations:**

The authors appropriately note that their evaluation is restricted to clean, 2D grid-based environments. Consequently, while the framework rigorously diagnoses structural perception errors in discrete spaces, extrapolating these specific failure modes to continuous, open-world multimodal tasks (such as robotic manipulation or complex UI automation) remains speculative and requires future empirical validation.

**Strengths And Weaknesses:**

Strengths:

S1. Diagnostic Verification Gating: The two-stage design is a highly principled approach. By exclusively evaluating reasoning accuracy on samples where the model's perception has already been verified, the authors effectively mitigate the common confound of visual hallucination masquerading as logical failure.

S2. Innovative Structural Error Analysis: Moving beyond binary accuracy, the quantification of coordinated off-by-one perception drift is a standout technical contribution. Using Monte Carlo baselines to isolate adjacency effects from marginal directional biases provides a deep, geometric understanding of how VLM perception fails.

S3. Clear Demonstration of the Simulation Gap: The matched-sample comparison (Equation 3) between Explicit (retrospective) and Predictive (prospective) modes is methodologically careful and convincingly demonstrates that VLMs struggle disproportionately with internal state simulation.

Weaknesses:

W1. Critical Absence of Text-Only/Symbolic Control Baselines: The paper concludes that the primary bottleneck is the "fundamental transformation of visual encoding into rule-based reasoning." However, this claim is unsubstantiated without a text-only control experiment. The authors must evaluate the underlying LLMs using purely symbolic representations (e.g., FEN strings for Chess or Xiangqi notation) on the exact same rule complexity ladder.  If the base LLM fails on the text-only input, the failure is a general logical reasoning deficit, not a multimodal visual-integration bottleneck.

W2. Confounding Variable in the Density Analysis: The authors interpret the counter-intuitive finding that "higher piece density improves perception accuracy" as evidence that feature density anchors spatial attention. However, in simulated gameplay, board density is heavily confounded with positional canonicity. High-density boards represent early/mid-game states (highly canonical and heavily represented in training data), whereas low-density boards represent endgames (often rare or unusual configurations). The observed accuracy drop may simply reflect the model's unfamiliarity with endgame distributions rather than a perceptual density issue.

W3. Overreaching Claims of "Causal Isolation" and Subset Bias: The paper repeatedly claims to "causally isolate" reasoning by conditioning on verified perception. This is methodologically inaccurate. Filtering by verified perception creates an observational, biased subset; as rule complexity increases, the subset of boards that perfectly pass verification may systematically shift toward easier or more canonical configurations. While the authors use a matched-sample intersection (Eq. 3) for the simulation gap, they fail to apply a similarly rigorous control for the main perception-reasoning dissociation claim (Figure 8).

W4. Lack of Prompting and Chain-of-Thought (CoT) Ablations: The predictive simulation tasks are evaluated strictly using zero-shot prompts. Predictive reasoning in complex board games typically requires multi-step planning. The massive "simulation gap" observed might largely be an artifact of zero-shot prompting. Without evaluating whether explicit Chain-of-Thought (CoT) or structured intermediate scratchpads (e.g., listing attacked squares first) can close this gap, asserting an architectural simulation limit is premature.

W5. Omission of Contemporary Diagnostic Literature and Baselines : This paper focuses heavily on general game-playing benchmarks but omits highly relevant contemporary works explicitly dedicated to diagnostic visual reasoning, disentanglement, and consistency. Including these in experiments and related works is necessary to properly position the paper's novelty:

[1] Alam et al. (2025), "SPHINX: A Synthetic Environment for Visual Perception and Reasoning".

[2] Just et al. (2025), "More Than the Final Answer: Improving Visual Extraction and Logical Consistency in Vision-Language Models".

[3] Wang et al. (2025), "Diagnosing the Compositional Knowledge of Vision Language Models from a Game-Theoretic View".

---

> ### Author Rebuttal · Authors · 2026-03-31
>
> We thank the reviewer for the constructive feedback.
>
> **W1/Q1: Text-only baseline.**
>
> Please see our response to Reviewer yc1i (W2) for full text-only baseline results and revision plan.
>
> **W2/Q2: Density analysis confound.**
>
> We thank the reviewer for this insightful critique. We conducted a controlled experiment to test the exact hypothesis: we re-ran the density evaluation on chess using entirely randomized piece placements, removing the correlation between density and positional canonicity.
>
> | Model | Low | Medium | High |
> | --- | --- | --- | --- |
> | Gemma-3 27B | 9.92 | 11.51 | 13.46 |
> | GPT-5.2 | 99.33 | 99.80 | 99.77 |
> | Qwen3-VL 8B | 80.70 | 71.42 | 72.58 |
> | Qwen3-VL 30B | 85.76 | 83.67 | 87.78 |
> | Qwen3-VL 235B | 96.21 | 95.42 | 95.91 |
>
> The results confirm the reviewer's hypothesis: when positional familiarity is controlled for, the clear upward density-accuracy trend in chess is no longer consistently present. We acknowledge that the original interpretation overstated the role of density as a perceptual factor in chess. The Gomoku results, which are free from this confound by design (Section 3.3.1, Appendix B.1), provide cleaner evidence for examining density effects.
>
> **Revision plan.** We will (1) replace the chess density results in Table 3 and Figure 3 with the random-placement data to remove the canonicity confound, (2) revise Section 4.1.1 to report that the density-accuracy trend is clearly observed in Gomoku, while in chess under controlled conditions the trend appears in some models but is inconsistent across the board, and (3) retain the original simulated-gameplay chess results in the appendix for transparency.
>
> **W3/Q3: Causal isolation claim and subset bias.**
>
> We agree that the term "causally isolates" is imprecise and will soften this wording in the revised manuscript.
>
> Cross-level matched-sample analysis (as in Equation 3) is not applicable here. L1–L6 test entirely different rules with independently generated samples—no case appears in more than one level. Equation 3 is applicable to the simulation gap because Explicit and Predictive modes evaluate the same samples with different question framing. The perception-reasoning dissociation, by contrast, is evidenced by the within-level gap between verification rate and reasoning accuracy.
>
> At any single level, this gap is already sufficient evidence. At L4, GPT-5.2 achieves 100% verification rate, yet reasoning accuracy is only 65%. Subset bias requires substantial filtering to be meaningful; when >98% of samples pass verification (GPT-5.2 L1–L5), bias has negligible room to operate.
>
> **W4/Q4: Chain-of-Thought ablation.**
>
> All models (except Gemma-3) already operate with native thinking mode enabled. To test whether additional explicit CoT helps, we augment the original prompts with:
>
> *"Analyze this step-by-step using the following scratchpad format: (1) State Identification: List the exact start and end coordinates of the moving piece. (2) Rule Retrieval: State the exact movement and capture rules for this specific piece type. (3) Trajectory Mapping: List every square the piece will pass through. (4) Constraint Checking: Check the start, path, and end squares against all rules and constraints identified in Step 2. (5) Final Conclusion: Based on the above, {question} (Yes/No/Unknown)"*
>
> We evaluate GPT-5.2 and Qwen3-VL 235B on L3–L6 matched samples.
>
>  [CoT vs Original](https://drive.google.com/file/d/1CWr2wm63hP3cNmi1zV2noDRDmaOwr_gH/view?usp=sharing)
>
> CoT does not reliably improve performance and frequently degrades it. For GPT-5.2, CoT yields gains in Predictive mode at L4–L5 (e.g., L5: 82.1% vs. 68.4%) but consistently underperforms in Explicit mode (e.g., L3: 81.3% vs. 96.5%). For Qwen3-VL 235B, CoT severely degrades Predictive L6 (23.1% vs. 41.7%) and Explicit L3 (77.1% vs. 95.5%).
>
> Additionally, the simulation gap persists under CoT, reaching 33.0pp at L6 (Explicit 85.9% vs. Predictive 52.9%)—comparable to or larger than without CoT.
>
> [Simulation Gap under CoT](https://drive.google.com/file/d/1k_cl6WKGAC4_Sat6a8IjmzMiVwY5vNd0/view?usp=sharing)
>
> This confirms the simulation gap is a structural limitation, not a prompting artifact: explicitly instructing VLMs to reason step-by-step does not resolve their inability to simulate unobserved visual states. We will include the full CoT ablation results in the revised appendix.
>
> **W5: Related work.**
>
> We will incorporate [1], [2], [3] into our Related Work. Briefly: [1] measures aggregate accuracy without attributing failures to perception vs. reasoning; [2] proposes decoupled training to separately fix perception and reasoning—complementary to our work which separately diagnoses them; [3] diagnoses compositional representations in dual-encoder VLMs, whereas we evaluate generative VLMs with verification gating.
>
> We appreciate the reviewer's rigorous critique, which has substantively improved our work. We hope these additions demonstrate the reliability and utility of our contributions.

---

> > ### Author Rebuttal · Reviewer_1cy3 · 2026-04-02
> >
> > Thank you for the detailed rebuttal and the additional experiments. I appreciate the authors’ efforts to strengthen the paper, especially through the added text-only baseline, the controlled random-placement experiment for the density analysis, and the explicit CoT ablation. I also appreciate the authors’ willingness to revise the overly strong causal language.
> >
> > I have also carefully read the other reviewers’ comments, and I particularly agree with Reviewer yc1i that the remaining issue is more fundamental than a missing ablation. While the added text-only baseline is helpful, it also makes the broader interpretation more delicate: if a substantial part of the observed failure can also be explained after replacing visual input with symbolic input, then the paper’s strongest conclusions about a specifically multimodal or architectural bottleneck require a more careful scope statement. More generally, I am not fully convinced that the paper’s decomposition into perception, rule reasoning, and simulation fully captures what should count as visual reasoning beyond this benchmark setting.
> >
> > Because this concern bears on the core framing and external validity of the work, and would likely require a more substantial revision of the paper’s positioning and claims, I keep my score unchanged.

---

### Official Review · Reviewer_5gD1 · 2026-03-10

**Soundness:** 3
**Presentation:** 2
**Significance:** 2
**Originality:** 2
**Overall Recommendation:** 3
**Confidence:** 4

**Summary:**

This paper proposes a two-stage diagnostic framework for decomposing Vision-Language Model (VLM) performance in game-based reasoning environments into independently testable components:*perception* and*rule-following*.


- The perception stage evaluates visual encoding through controlled tests manipulating density, patch alignment, resolution divisibility, and visual richness across chess (8×8) and Gomoku (15×15).

- The rule-following stage introduces a 2×2 diagnostic matrix (single-state/multi-state × rule-free/rule-based) and a six-level rule complexity ladder, each evaluated in Explicit (verification) and Predictive (simulation) modes, on chess and Xiangqi.


The rule of reasoning tests is basically: hold perception fixed, then increase rule complexity in a controlled way and see where reasoning collapses.


Experiments on 6 VLMs (Qwen3-VL 8B/30B/235B, GPT-5.2, GLM-4.1V 9B, Gemma-3 27B) yield three findings (from the abstract):

1. Spatially coordinated localization drift, where adjacent pieces exhibit same-direction off-by-one errors at 1.5–1.9× the rate expected under independence

2.  Perception-reasoning dissociation, where models correctly verify board states but fail to apply rules, with reasoning accuracy declining steeply as complexity increases even when perception remains stable

3.  A simulation gap of up to 27pp between Explicit verification and Predictive simulation modes.



The authors argue that these limitations persist across model scales and architectures, **suggesting** a structural limitation in current VLMs (especially in converting visual perception into rule-based reasoning) rather than a problem that scaling alone can fix.

**Compliance With Llm Reviewing Policy:**

Affirmed.

**Final Justification:**

After reading the rebuttal and the reviewer discussion, I will keep my score unchanged. I appreciate the added text-only baseline and the clearer positioning against prior work, which strengthen the paper. However, my main concern remains unresolved: I am still not convinced that the proposed decomposition captures visual reasoning beyond this benchmark setting, since much of the task appears reducible to state extraction followed by symbolic rule application.

**Key Questions For Authors:**

1. How does your diagnostic framework concretely advance beyond LVLM-Playground (Wang et al., 2025a), which already decomposes VLM game performance into Perceiving, Rule Following, and other stages?

2. Community does know VLMs struggle in  translating detailed visual features into words, including reading board configuration. Beyond this, what is the most interesting finding from your experiments (in the percpetion)?

3. Could you provide provide text-only baselines to compare against VLMs results?

4. The Qwen3-VL family shares vision encoder architecture, making it difficult to attribute spatial drift patterns to **universal** architectural phenomena versus family-specific encoder biases. Can you provide evidence that the directional preferences reflect general VLM properties rather than encoder behaviors?

5. Can visual prompting methods such as augmenting various grids help the perception tasks (appending rule like corners of image, or overlaying grid on top of the image)?

**Limitations:**

Partly yes.  however some important limitations are not discussed:

- The heavy reliance on the Qwen3-VL family (three of six models)

- Chess and Gomoku are heavily represented in LLM/VLM training data and could to elevated numbers.

- No discussion of long CoT / inference scaling.

**Strengths And Weaknesses:**

## Strengths

- Spatial drift analysis is interesting - The structural shift analysis is the paper's most distinctive empirical contribution (the use of Hungarian matching with Chebyshev distance for piece alignment).

- Comprehensive appendix and reproducibility - The appendix is thorough, providing complete prompt templates, board generation procedures, patch alignment configurations, resolution divisibility specifications, and rule complexity ladder definitions at all six levels.


## Weaknesses

- Originality and Significance: Prior work has shown limitations of VLMs in perception [1,2,3,4] across various setups, and LVLM-Playground [3] (Wang et al., cited in related work) already factors VLM board game performance into four task categories: Perceiving, Question Answering, Rule Following, and End-to-End Playing. The paper does not adequately position itself with respect to many prior work, see [1,2,3,4].

-  Comparisons do not tell us whether the *reasoning* weakness is multimodal-specific or just a general reasoning weakness the same model would also have in text form.  Without a *text-only baseline*, the paper cannot cleanly claim VLMs are bad/good at reasoning. (The paper's reasoning tasks are designed so that, once the visual state/history has been correctly extracted, the rest of the task can be carried out as symbolic rule reasoning).


### References

1. Rahmanzadehgervi, Pooyan, et al. "Vision language models are blind." Proceedings of the Asian Conference on Computer Vision. 2024.
2. Chia, Yew Ken, et al. "Puzzlevqa: Diagnosing multimodal reasoning challenges of language models with abstract visual patterns." Findings of the Association for Computational Linguistics: ACL 2024. 2024.
3. Wang, Xinyu, Bohan Zhuang, and Qi Wu. "Are large vision language models good game players?." arXiv preprint arXiv:2503.02358 (2025).
4. Yan, Hao, et al. "Visuriddles: Fine-grained perception is a primary bottleneck for multimodal large language models in abstract visual reasoning." arXiv preprint arXiv:2506.02537 (2025).

---

> ### Author Rebuttal · Authors · 2026-03-31
>
> We thank the reviewer for the constructive feedback.
>
> **W1/Q1: Positioning with respect to prior work.**
>
> We studied LVLM-Playground (Wang et al., 2025a) extensively in the early stages of this work and identified several limitations that directly motivated our framework design.
>
> At the evaluation level, LVLM-Playground's rule-following test asks models to enumerate all legal moves from a given board state. This conflates perception and reasoning into a single aggregate score and, due to the large number of possible legal moves, allows partial matches by chance—making it difficult to assess genuine rule understanding. Our framework addresses this through (1) verification gating that separates perception from reasoning, and (2) a rule complexity ladder that isolates specific rule types at controlled difficulty levels rather than testing all legal moves jointly.
>
> At the data level, during reproduction we found incorrect coordinate axis labeling in Gomoku prompts, output parsing errors that misclassify valid responses containing blank lines as failures, illegal board states in Chess samples, and unfiltered degenerate outputs (one model outputs identical responses across all 2,000 samples). These issues further motivated the development of a more controlled framework with verified board generation and robust output parsing, as detailed in Appendix A.3.
>
> Regarding the other references: [1] demonstrates VLMs fail at basic spatial perception, consistent with our findings, though it does not examine whether reasoning succeeds when perception is correct. [2] is most related—PuzzleVQA also separates perception from reasoning, but relies on oracle hints to diagnose bottlenecks, whereas our verification gating requires no ground-truth assistance.  [4] similarly identifies fine-grained perception as the primary bottleneck in abstract visual reasoning and proposes a training-based solution, complementing our diagnostic focus. We will add all four references to the revised Related Work.
>
> **W2/Q3: Text-only baseline.** Please see our response to Reviewer yc1i (W2).
>
> **Q2: Most interesting perception finding.**
>
> Our most distinctive finding is the spatially coordinated localization drift (Section 4.1.5): adjacent pieces shift in the same direction at 1.5–1.9× the rate expected under spatial independence, verified against Monte Carlo baselines. This goes beyond reporting that VLMs make perception errors—it reveals that these errors have structured spatial geometry, with localized board regions displaced as a unit. This pattern has not been documented in prior work and has practical implications for domains like document understanding or UI automation, where spatial errors may propagate as regional displacements rather than independent noise.
>
> Our negative findings that patch alignment, resolution divisibility, and visual richness have minimal impact on perception accuracy—may help guide future research by suggesting that these factors are less promising directions for understanding VLM perception failures in grid-based settings.
>
> **Q4: Whether spatial drift is Qwen-family-specific.**
>
> The three non-Qwen models use different vision architectures: Gemma-3 27B uses a frozen SigLIP-400M encoder (Kamath et al., 2025), GLM-4.1V uses an independently trained ViT encoder with MLP projection (Hong et al., 2025), and GPT-5.2 (undisclosed, reportedly natively multimodal). All six models exhibit the effect (Figures 5–6). GPT-5.2 shows the strongest structural coordination (1.9× baseline), while GLM-4.1V and Gemma-3 also exhibit coordinated shifts at 1.5× and 1.7× their respective baselines. Each model also shows distinct directional preferences (Figure 7), indicating independent encoding biases.
>
> **Q5: Whether visual prompting can help perception.**
> Our board images already include grid lines and coordinate labels (Figure 12, Appendix B). Since coordinated spatial drift persists despite these explicit visual anchors, additional overlays are unlikely to resolve the phenomenon and may introduce visual noise.
>
> **Limitation 1: Qwen3-VL reliance.** We acknowledge that three of six models share the Qwen3-VL architecture. This choice was deliberate: testing three scales (8B, 30B, 235B) within one family enables controlled analysis of scaling effects (Section 4.2.4). As shown in Q4, all six models exhibit coordinated spatial drift, and GPT-5.2 shows the same rule-following phenomena, confirming our findings are not Qwen-specific.
>
> **Limitation 2: Training data familiarity.** This is a valid concern, but the selection of well-known games is deliberate: our framework tests whether models can *apply* rules they already know, not learn unfamiliar ones from prompt. As verified in Appendix A.2.3, all models demonstrated correct rule knowledge. We include Xiangqi, less represented in English training data, to test cross-domain generalization—the same failure patterns persist.
>
> **Limitation 3: Long CoT.** Please see our response to Reviewer 1cy3 (W4).

---

> > ### Author Rebuttal · Reviewer_5gD1 · 2026-04-03
> >
> > I thank the authors for their rebuttal. While I appreciate the authors' efforts to resolve some of the issues, I agree with the other two reviewers’ comments that the main remaining concern is the paper's core framing: I am not fully convinced that the proposed decomposition reflects genuine visual reasoning beyond this benchmark setting.

---

### Official Review · Reviewer_gGnQ · 2026-03-13

**Soundness:** 4
**Presentation:** 4
**Significance:** 3
**Originality:** 4
**Overall Recommendation:** 4
**Confidence:** 3

**Summary:**

This paper analyzes failure modes of Vision–Language Models (VLMs) in game-based reasoning environments. The authors propose a diagnostic framework that decomposes gameplay performance into perception, rule-based reasoning, and simulation, and evaluate six VLMs. The study identifies three key phenomena: coordinated spatial drift, perception–reasoning dissociation, and a simulation gap between verification and prediction tasks, providing insights into limitations of current VLM systems.

**Compliance With Llm Reviewing Policy:**

Affirmed.

**Key Questions For Authors:**

- The framework is mainly evaluated in game environments. To what extent do the identified failure patterns (e.g., spatial drift and simulation gaps) generalize to other multimodal reasoning tasks such as document understanding, robotics perception, or visual planning?
- The paper observes coordinated spatial drift across nearby objects. Have the authors investigated whether this phenomenon originates from the visual encoder, the tokenization process, or downstream reasoning modules?
- Many VLM reasoning abilities depend heavily on prompting strategies. Have the authors tested whether chain-of-thought or structured prompting can mitigate the perception–reasoning dissociation or simulation gap observed in the experiments?
- The experiments introduce a ladder of rule complexity. How sensitive are the results to the specific definition of these complexity levels? Would different rule structures lead to similar failure patterns?

**Limitations:**

Yes.

The authors acknowledge several limitations, including the focus on specific game environments and the diagnostic nature of the framework rather than proposing new model architectures. The paper appropriately positions its contribution as analysis and evaluation rather than algorithmic improvement.

**Strengths And Weaknesses:**

Strengths

1. The paper highlights an important limitation of current VLM benchmarks: most evaluations only report overall accuracy and do not reveal the underlying causes of model failures. By separating perception, reasoning, and simulation, the work focuses on error attribution, which is useful for understanding multimodal reasoning systems.
2. The proposed two-stage diagnostic framework is well designed. The combination of controlled perception tests and a diagnostic matrix over rule complexity and reasoning modes provides a systematic way to analyze model behavior and distinguish different types of errors.
3. The empirical analysis uncovers several interesting patterns, including coordinated spatial drift and the performance gap between verification and prediction tasks. These findings offer useful insights into how VLMs process visual information and perform reasoning under structured constraints.
4. Although the experiments are conducted in game environments, the paper reasonably argues that such environments serve as controlled testbeds for multimodal reasoning and decision-making. The observations may therefore have implications for related domains such as robotics, UI automation, and document understanding.

### Weaknesses

1. The methodological novelty is somewhat limited. The main contribution lies in the diagnostic framework and empirical analysis rather than in proposing a new learning algorithm or modeling technique. Compared with typical ICML papers that introduce new architectures, paradigms, or theoretical insights, the novelty is relatively modest.
2. While the paper reports interesting empirical phenomena (e.g., spatial drift and simulation gaps), it does not provide deeper mechanistic explanations. It remains unclear whether these failures originate from representation limitations, prompting strategies, training data biases, or architectural constraints.
3. The experimental scope is somewhat narrow. Although six recent VLMs are evaluated, the experiments focus mainly on board-style game environments. It would strengthen the work to investigate whether the observed failure modes also appear in other structured visual reasoning tasks.
4. The paper focuses on diagnosing failure modes but does not explore possible mitigation strategies. Including preliminary experiments or design suggestions could further increase the practical impact of the work.

---

> ### Author Rebuttal · Authors · 2026-03-31
>
> We thank the reviewer for the constructive feedback.
>
> **W1: Methodological novelty.**
>
> We agree that our contribution is primarily diagnostic rather than algorithmic. We believe this is appropriate and valuable: as noted in our Related Work (Section 2.2), existing benchmarks report *that* VLMs fail but rarely diagnose *why*. Our framework fills this gap by enabling fine-grained failure attribution—a prerequisite for designing effective solutions. The new text-only and CoT ablations (see our responses to Reviewers yc1i W2 and 1cy3 W4) further demonstrate the framework's utility as a diagnostic tool: they reveal that reasoning failures arise from both general and multimodal-specific limitations, and that structured prompting does not reliably mitigate them. These are actionable insights that can guide future architectural and training improvements.
>
> **W2: Lack of mechanistic explanations.**
>
> We agree that our paper focuses on characterizing failure patterns rather than identifying their mechanistic origins, and we view reliable diagnosis as a necessary precursor to mechanistic investigation. That said, our new ablations help narrow the possible causes: the CoT ablation (see our response to Reviewer 1cy3 W4) rules out prompting strategy as the primary factor, and the text-only baseline (see our response to Reviewer yc1i W2) shows that reasoning failures persist without visual input, indicating that general reasoning limitations also contribute beyond architectural bottlenecks. Deeper mechanistic analysis—such as probing internal representations or analyzing attention patterns—is an important direction for future work.
>
> **W3: Narrow experimental scope.**
>
> We agree that extending to non-grid structured tasks (e.g., document parsing, circuit diagrams, UI navigation) is a valuable direction. Our choice of board games is deliberate: they provide fully specified rules, unambiguous state transitions, and precise ground truth—properties necessary for the controlled diagnostic methodology we propose. As discussed in our response to Reviewer yc1i (W1), the failure modes we identify are relevant to any structured domain where accurate state recognition must be reliably coupled with rule application. Extending the framework to additional domains is a natural next step.
>
> **W4: No exploration of mitigation strategies.**
>
> Our CoT ablation (see our response to Reviewer 1cy3 W4) is a first step in this direction—it tests whether structured prompting can mitigate the observed failures. The result that CoT does not reliably help (and often hurts) is itself informative for practitioners, as it suggests that simple prompting-based mitigations are insufficient. We agree that exploring more targeted strategies (e.g., perceptual fine-tuning, decoupled training as in PeRL-VL) is important future work.
>
> **Q1: Generalizability to other multimodal reasoning tasks.**
>
> Our framework is directly applicable to any structured domain where perception and rule application can be decomposed—such as document understanding, UI automation, and visual planning. We discuss this in detail in our response to Reviewer yc1i (W1).
>
> **Q2: Origin of spatial drift.**
>
> Our patch alignment test (Section 4.1.2) shows that models are highly robust to where elements fall relative to tokenization boundaries, suggesting that the tokenization process itself is not the source. Combined with the spatially coordinated nature of the drift—adjacent pieces shifting together rather than exhibiting random semantic confusion—this points toward the visual encoder's spatial grounding as the more likely bottleneck. We acknowledge this remains a hypothesis rather than a confirmed mechanism, and will add this discussion to the revised manuscript.
>
> **Q3: Whether structured prompting can mitigate the failures.**
>
> Please see our response to Reviewer 1cy3 (W4).
>
> **Q4: Sensitivity to rule complexity definitions.**
>
> Our complexity ladder spans two different games (Chess and Xiangqi) with fundamentally different rule systems—Chess involves en passant, castling, check constraints, etc., while Xiangqi involves flying general, cannon screen captures, perpetual check/chase detection, etc. Despite these entirely different rule structures, the same qualitative patterns emerge: reasoning accuracy declines with complexity while perception remains stable, and the Explicit–Predictive gap widens at higher levels. This consistency across two independent rule systems suggests that the observed failure patterns are not artifacts of our specific complexity definitions but reflect general limitations in multi-constraint rule application.

---

> > ### Author Rebuttal · Reviewer_gGnQ · 2026-04-03
> >
> > My questions are properly addressed.

---

### Official Review · Reviewer_yc1i · 2026-03-13

**Soundness:** 2
**Presentation:** 3
**Significance:** 2
**Originality:** 3
**Overall Recommendation:** 3
**Confidence:** 4

**Summary:**

The paper introduces an analysis framework for diagnosing VLMs by disentangling errors in visual understanding, rule-based reasoning, and forward-state prediction. The study shows that these models exhibit patterned perceptual errors, with multiple neighboring pieces often displaced in the same direction. It also finds a strong separation between seeing and reasoning: models can accurately read the board state yet still struggle once the game’s rules demand more complex inference. In addition, the authors identify a consistent weakness in simulation, showing that models are far better at checking whether a given result matches the current board than at forecasting valid future moves. Larger models do not consistently overcome this limitation.

**Compliance With Llm Reviewing Policy:**

Affirmed.

**Final Justification:**

My concern has not been fully addressed. I decide to maintain my original score.

**Key Questions For Authors:**

Please see Weaknesses.

**Limitations:**

yes

**Strengths And Weaknesses:**

Strengths: the authors provide a diagnosing VLMs by disentangling errors in visual understanding, rule-based reasoning, and forward-state prediction and then reveals some intriguing findings in these aspects. Paper are well presented with figures and easy to understand.

Weaknesses:
1. The most significant weakness is that this framework and analysis do not generalize to the genuine visual reasoning process, where visual and text information need to be integrated for reasoning. This paper analyzes the game play as a perception, then reasoning process where viusal perception is only used to identify or translate an image into a set of states, while in the real world, problems such as spatial reasoning (perspective taking, etc) cannot be dissociated as perception and then a reasoning process. This limits this work from studying a genuine visual reasoning process.
2. The game states can be directly given to LLMs, which can serve as a text-only baseline.
3. Most findings rely on observing accuracy gaps and interpreting them post-hoc as one failure mode or another. More convincing would be designed interventions — e.g., injecting corrected perceptions mid-task or ablating rule complexity — to causally isolate the three failure types rather than inferring separation indirectly.

---

> ### Author Rebuttal · Authors · 2026-03-31
>
> We thank the reviewer for the constructive feedback.
>
> **W1: Generalizability to genuine visual reasoning.**
>
> We agree that our framework assumes decomposability of perception and reasoning, and that this assumption does not hold for all visual reasoning tasks—perspective taking and embodied spatial reasoning, as the reviewer notes, are examples where perception and reasoning are more tightly intertwined. We acknowledge this as a genuine scope limitation.
>
> However, we respectfully note that many practical multimodal tasks *are* well characterized by a perception-then-reasoning pipeline: document understanding (perceive layout and text → apply extraction rules), UI automation (perceive interface elements → follow interaction logic), and visual planning in structured environments (perceive state → apply constraints). Our paper explicitly positions games as controlled testbeds for these structured domains (Section 1), not as end applications. The failure modes we identify are directly relevant to any structured domain where accurate state recognition must be reliably coupled with rule application. Specifically, the coordinated spatial drift suggests that perception error mitigation cannot assume independent noise but must account for structured regional displacements; the perception-reasoning dissociation indicates that verifying visual input alone is insufficient without independent validation of downstream reasoning; and the simulation gap cautions that verification-based evaluations may substantially overestimate a model's ability to perform predictive reasoning in deployment.
>
> We will clarify the scope of our framework in the revised Limitations section.
>
> **W2: Text-only baseline.**
>
> We evaluate GPT-5.2 and Qwen3-VL 235B on our rule complexity ladder (L3–L6) in a text-only setting, replacing image input with FEN strings while keeping the reasoning question identical. Accuracy is computed only on cases where the image-based run passed verification, so both conditions are evaluated on the exact same test instances.
>
> [Text-Only vs Original](https://drive.google.com/file/d/15ABND2cDkI0B3L57q-eK8-ejR9Bz7PsY/view?usp=sharing)
>
> The text-only baseline does not uniformly outperform image-based verified reasoning. At L3, image-based reasoning frequently matches or exceeds text-only (e.g., GPT-5.2 Predictive: image 98.5% vs. text 83.9%). At higher complexity, text-only often outperforms (e.g., Qwen3-VL 235B Predictive L6: text 79.0% vs. image 41.7%), but not universally—GPT-5.2 Predictive L6 shows image slightly ahead (75.2% vs. 70.6%).
>
> By introducing this text-only intervention, we can disentangle baseline symbolic reasoning capacity from the overhead introduced by the visual modality. While text-only reasoning also degrades at high complexity—confirming that general reasoning limitations contribute—the image-based pathway suffers a distinct, quantifiable additional cost in most conditions (e.g., Qwen3-VL 235B Predictive L6: 79.0% text vs. 41.7% image). This strengthens rather than weakens our core finding: the multimodal processing pipeline introduces reasoning overhead that persists even when perception is verified correct.
>
> **Revision plan.** In the revised manuscript, we will (1) add the text-only baseline as a new subsection, extending to all models with sufficient verification rates and covering L1–L6, with per-game breakdowns in the appendix, (2) update the Limitations section to reflect that text-only baselines have been addressed, (3) revise our causal attribution throughout the paper (Abstract, Sections 4.2.2, 4.2.4, and 5) to reflect that the observed reasoning failures arise from both general reasoning limitations and multimodal transformation overhead, rather than attributing them solely to the visual-to-symbolic transformation pipeline.
>
> **W3: Designed interventions vs. post-hoc observation.**
>
> We acknowledge that verification gating is an observational filter rather than an active causal intervention. However, our text-only baseline above provides precisely the type of intervention the reviewer requests: by completely replacing visual input with symbolic representations and testing the same rule complexity ladder, we ablate the visual modality and isolate the reasoning mechanism from the visual processing pipeline. The results confirm that reasoning failures arise from both general reasoning limitations and multimodal transformation overhead, refining our original attribution. Additionally, our CoT ablation (see our response to Reviewer 1cy3, W4) constitutes a second intervention that manipulates prompting strategy while holding visual input constant. Together with the controlled variations already in the framework (rule complexity ladder, Explicit vs. Predictive mode contrast), these interventions move our analysis beyond post-hoc interpretation.

---

> > ### Author Rebuttal · Reviewer_yc1i · 2026-04-01
> >
> > Thanks for your rebuttal. While I appreciate the added experiments with the text-only baseline, the rest of the concern remains unresolved and is critical to the proposed framework. Specifically, I believe it's hard to argue that the kind of visual reasoning that can decompose into perception and text reasoning is a genuine visual reasoning process, since this suggests that you can perform visual reasoning simply with a captioner and a reasoning LLM. Under the light of this, it's not even worth studying VLM or MLLM anymore. However, we know this is not the case since there are so many more visual problems that are beyond the simple caption + reason process.
> >
> > Best
> > reviewer

---

### Decision · Program_Chairs · 2026-04-30

**Decision:**

Accept (regular)

**Comment:**

This paper proposes a two-stage diagnostic framework to pinpoint why VLMs fail at game-based reasoning, rather than just reporting that they do. Most existing benchmarks conflate perception errors with reasoning failures into a single accuracy score; this work pulls them apart. The first stage runs controlled perception tests to isolate visual encoding ability. The second stage, assuming correct perception, evaluates rule-following through a six-level complexity ladder in two modes and predictive simulation. Some Reviewers gave Weak Reject; if the task can be decomposed into perception followed by symbolic reasoning, does it really test genuine visual reasoning. The authors pushed back with new text-only baselines during rebuttal.